# Learning Arbitrary Logical Formula as a Sparse Neural Network Module

## Abstract

NeSy (Neuro-Symbolic) predictors are hybrid models composed of symbolic predictive models chained after neural networks. Most existing NeSy predictors require either given symbolic knowledge or iterative training. DSL (Deep Symbolic Learning) is the first NeSy predictor that supports fully end-to-end training from scratch, but it learns a look-up table rather than arbitrary programs or formulas. We propose the Logical Formula Learner framework, a general framework of network modules that explicitly equate a logical formula after convergence. We then propose 3 novel designs within the LFL framework with different levels of combinatorial search freedom: LFL-Type1 learns arbitrary logical formula, LFL-Type2 learns a look-up table, and LFL-Type3 has combinatorial search freedom between them. LFL-Type1 and LFL-Type2 show improvements over previous designs, and all three types can be wrapped into NeSy predictors. To our knowledge, LFL-Type1-based NeSy predictor is the first NeSy predictor that supports fully end-to-end training from scratch and explicitly learns arbitrary logical formulas. [1]

## 1 Introduction

In a special lecture of NIPS 2019, Yoshua Bengio suggested that studies in deep learning should expand from System 1 to System 2 (Bengio, 2019), where System 1 means the intuitive, black-box parts and System 2 means the symbolic, white-box parts of either human or artificial intelligence. Works in multiple research directions can be seen as attempting to approach System 2 intelligence, such as those combining deep learning with Symbolic Regression (SR) (Makke & Chawla, 2024) or Inductive Logic Programming (ILP) (Zhang et al., 2023) methods, while the term Neuro-Symbolic (NeSy) refers to the general research field of combining neural systems and symbolic reasoning (Marra et al., 2024).

A subset of NeSy methods is NeSy predictors defined in Marconato et al. (2024), members of which present hybrid models composed of symbolic predictive models chained after neural networks. Some NeSy predictors combine neural networks with a given symbolic program or knowledge base so that the neural network can be trained by symbolic constraints (Manhaeve et al., 2018; Huang et al., 2021; Badreddine et al., 2022; Winters et al., 2022; Yang et al., 2023). Some other NeSy predictors attempt to learn the symbolic predictor jointly with the neural networks, most of them requiring iteratively training the symbolic predictor and the neural networks while fixing the other (Duan et al., 2022; Cunnington et al., 2022; Liu et al., 2023); some of them also require pre-training the neural networks beforehand. Then DSL (Deep Symbolic Learning) (Daniele et al., 2022) presents a fully differentiable NeSy predictor that allows for end-to-end training of a differentiable logic module chained after neural networks, both from scratch. The major limitation of DSL is that its differentiable logic module equates a look-up table that maps each of the input concepts' cartesian product to one of the possible output symbols, while many other NeSy predictors learn arbitrary logical circuits or programs.

The idea of designing a differentiable module that becomes equivalent to a symbolic expression is also seen in a line of research starting from EQL (Equation Learner) (Martius & Lampert, 2016), a differentiable module designed for SR. An EQL network is a neural network module that equates an arithmetic formula after convergence, archived by introducing diverse activation functions and

---

[1]Code available at: https://anonymous.4open.science/r/logical-formula-learner-693B

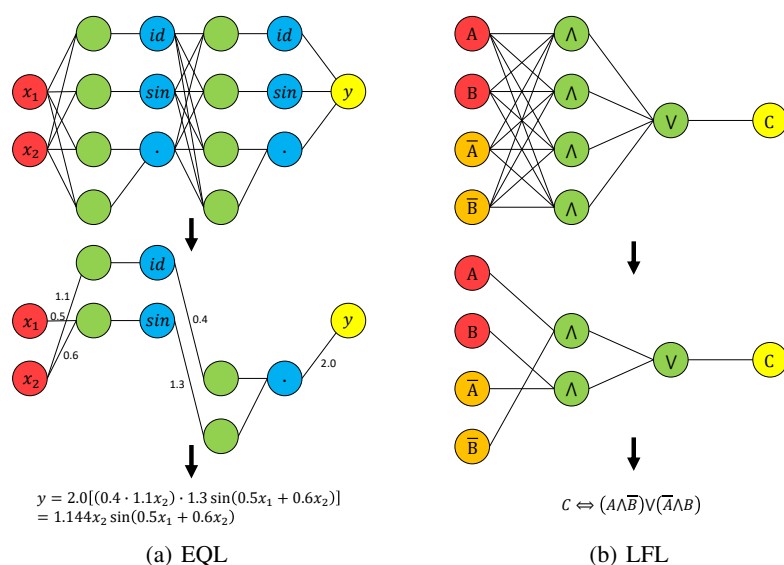

$$y = 2.0[(0.4 \cdot 1.1x_2) \cdot 1.3 \sin(0.5x_1 + 0.6x_2)]$$
$$= 1.144x_2 \sin(0.5x_1 + 0.6x_2)$$

$$C \Leftrightarrow (A \wedge \overline{B}) \vee (\overline{A} \wedge B)$$

(a) EQL                    (b) LFL

Figure 1: An intuitive illustration of how an EQL or LFL converges into symbolic expressions. In all network architecture figures of this paper, red-colored objects represent input values, yellow ones represent output values, orange ones represent intermediate values, green ones represent trainable modules or neurons, and blue ones represent transformations that contain no trainable parameters.

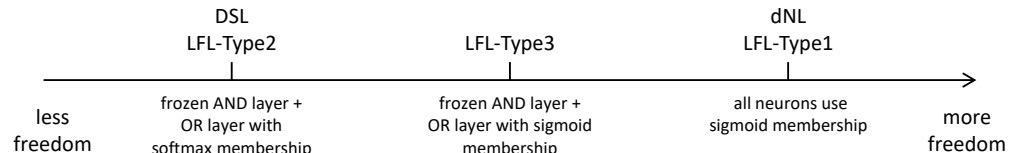

Figure 2: Comparison of LFL variations in terms of freedom of their combinatorial search spaces.

sparsity constraints into an MLP network. Another method with a similar philosophy is dNL (differentiable Neural Logic Network) (Payani & Fekri, 2019a). A dNL network is composed of logical neurons designed with Product t-norm that express AND or OR relationships and thus can be equivalent to any arbitrary logical formula, experimented mainly on ILP tasks.

Inspired by them, in this work we propose a general framework of differentiable modules that equate logical formulas after convergence, named Logical Formula Learner (LFL). An LFL network uses customized neurons that relax binary logic symbols with fuzzy logic operations and converges into a sparse network equating a relatively simple logical formula. Figure 1 shows an intuitive comparison between EQL and LFL.

Both DSL and dNL fit into the LFL framework. Apart from them, we propose three novel designs within the LFL framework with different levels of combinatorial search freedom (Figure 2): LFL-Type1 learns arbitrary logical formulas like dNL, LFL-Type2 learns a look-up table like DSL, and LFL-Type3 has combinatorial search freedom between them. LFL-Type1 works a little better than dNL on learning logical formulas from binary data when the number of hidden neurons is limited. LFL-Type2 and LFL-Type3 work as well as DSL's logic module in NeSy predictors with better consistency between training and inference behaviours. Although directly using LFL-Type1 in NeSy predictors makes the network struggle to converge, we tackle this problem by adding an MLP as gradient shortcut. This, to our knowledge, results in the first NeSy predictor that satisfies: (1) Being end-to-end differentiable; (2) Training all modules from scratch and achieving joint convergence in

a single run; (3) Explicitly learning arbitrary logical formula (within limited complexity) with its symbolic module.

## 2 THE LOGICAL FORMULA LEARNER FRAMEWORK

### 2.1 RELAXING BINARY LOGICAL NEURONS INTO FUZZY, DIFFERENTIABLE ONES

It is straightforward to define binary neurons that describe the logical AND, OR or NOT relationships:

$$f_{binary\_AND}(\boldsymbol{x}^n) = \bigwedge_i (\overline{m_i} \vee x_i) \tag{1}$$

$$f_{binary\_OR}(\boldsymbol{x}^n) = \bigvee_i (m_i \wedge x_i) \tag{2}$$

$$f_{binary\_NOT}(x) = \overline{x} \tag{3}$$

where $f$ means the neuron's output, $x_i \in \{0, 1\}$ means the neuron's $i$th input, and $m_i \in \{0, 1\}$ means the "membership" of the input which decides whether the input is a member of the AND/OR operation. The NOT operation applies on a single input.

To make our logical neurons differentiable and trainable, we need to replace the binary logical operations with fuzzy ones and the binary memberships with continuous values in range $[0, 1]$:

$$f_{AND}(\boldsymbol{x}^n) = \bigotimes_i ((1 - g(w_i)) \oplus x_i) = \bigotimes_i (1 - g(w_i) \otimes (1 - x_i)) \tag{4}$$

$$f_{OR}(\boldsymbol{x}^n) = \bigoplus_i (g(w_i) \otimes x_i) = 1 - \bigotimes_i (1 - g(w_i) \otimes x_i) \tag{5}$$

$$f_{NOT}(x) = 1 - x \tag{6}$$

where $f$ means the neuron's output, $x_i \in [0, 1]$ means the neuron's $i$th input, $\otimes$ and $\bigotimes$ mean chosen differentiable fuzzy t-norm operations, $\oplus$ and $\bigoplus$ mean chosen differentiable fuzzy t-conorm operations, $w_i$ means the control parameters for the membership value, and $g$ means a chosen differentiable function that maps $w_i$ into $[0, 1]$. An LFL network can be constructed by arbitrarily combining these differentiable neurons. The selected $g$ may produce soft, stochastic membership values during training. During inference, we set the memberships to binary, deterministic values in $\{0, 1\}$ according to the trained parameters $w_i$ so that the LFL module strictly equates a fuzzy logical formula.

### 2.2 HOW EXISTING METHODS FIT INTO THE LFL FRAMEWORK

#### 2.2.1 DIFFERENTIABLE NEURAL LOGIC NETWORK (DNL)

The logical neurons proposed in dNL can be obtained by substituting Product t-norm and the corresponding t-conorm as chosen fuzzy logic operations and the sigmoid function $\sigma$ as $g$ into equations 4 and 5 [2]:

$$f_{dNL\_AND}(\boldsymbol{x}^n) = \prod_i (1 - \sigma(w_i)(1 - x_i)) \tag{7}$$

$$f_{dNL\_OR}(\boldsymbol{x}^n) = 1 - \prod_i (1 - \sigma(w_i) x_i) \tag{8}$$

A dNL network can thus be constructed by arbitrarily combining the AND/OR neurons defined above and the NOT neurons defined in 6.

---

[2]The original design in dNL multiplies $w_i$ with a hyperparameter $c$, i.e. $g(w_i) = \sigma(cw_i)$. The hyperparameter $c$ seems redundant since scaling the trainable control parameters is equivalent to scaling the learning rate.

### 2.2.2 DEEP SYMBOLIC LEARNING (DSL)

As a fully differentiable NeSy predictor, a DSL network consists of black-box neural classifiers that map high-dimensional input to symbolic representations and a differentiable logic module that equates a look-up table after convergence, indicating that the logic module should also fit into the LFL framework. In this subsection, we describe an LFL module that is equivalent to the original definition of DSL's logic module.

Say there are multiple neural classifiers with their number of classes denoted as $\{n_1, n_2, n_3, ...\}$ and the $j$th unnormalized prediction of the $i$th classifier as $N_{ij} \in \mathbb{R}$. Each classifier's predictions are sparsified by $\epsilon$-greedy policy:

$$x_k = \begin{cases} \text{softmax}_j \left( N_{ij} \right), & \text{if } j \text{ is the chosen symbol of classifier } i \\ 0, & \text{otherwise} \end{cases} \tag{9}$$

where $k \in [1, \sum_i n_i]$, $\text{softmax}_j$ means applying softmax on the $j$ dimension, and $x_k \in (0, 1)$ is the LFL module's $k$th input. For each classifier, its predicted class is chosen with probability $1 - \epsilon_1$, and a random class is chosen with probability $\epsilon_1$.

The LFL module uses Gödel t-norm and t-conorm. Its first layer consists of AND neurons with memberships frozen such that each AND neuron represents a member of the cartesian product of the module's input concepts:

$$h_l = \min_k \left( 1 - \min \left( m_{kl}, 1 - x_k \right) \right) \tag{10}$$

$$m_{kl} = \begin{cases} 1, & \text{if the } k\text{th symbol is a member of the } l\text{th cartesian product of all class symbols} \\ 0, & \text{otherwise} \end{cases} \tag{11}$$

where $l \in [1, \prod_i n_i]$, $m_{kl}$ is the $l$th AND neuron's $k$th membership value, and $h_l$ is the neuron's output.

The second layer consists of OR neurons with trainable memberships generated with softmax and $\epsilon$-greedy policy, producing the final class prediction from $n_p$ classes:

$$y_p = \max_l \left( \min \left( m_{lp}, h_l \right) \right) \tag{12}$$

$$m_{lp} = \begin{cases} \text{softmax}_p \left( w_{lp} \right), & \text{if } p \text{ is the chosen symbol of input } l \\ 0, & \text{otherwise} \end{cases} \tag{13}$$

where $p \in [1, n_p]$, $m_{lp}$ is the $p$th OR neuron's $l$th membership value, $w_{lp}$ is the trainable control parameter for $m_{lp}$, $\text{softmax}_p$ means applying softmax on the $p$ dimension, and $y_p$ is the neuron's output. For each input $h_l$, its maximal $m_{lp}$ is chosen with probability $1 - \epsilon_2$, and a random one of the memberships is chosen with probability $\epsilon_2$.

During each individual input sample's forward pass, the above design assures that only one value in $x$, $h$ or $y$ is non-zero, so that the original definition of DSL could focus only on the non-zero values. The non-zero value of $y$ then defines the network's predicted class and truth value. Furthermore, the module can predict multiple class labels by concatenating outputs of multiple OR layers defined above.

### 2.3 OUR NEW DESIGNS

One difference between dNL and DSL is that the latter uses noisy input symbols and memberships with Gödel t-(co)norm, which may allow the network to explore more of the possible combinatorial search space. We also notice that the Concrete distribution (Maddison et al., 2016) provides an elegant way of adding noise to soft symbolic values. So in this subsection, we introduce three novel designs within the LFL framework, all of which utilize Gödel t-(co)norm and the Concrete distribution.

### 2.3.1 LFL-TYPE1: LEARNING ARBITRARY LOGICAL FORMULA LIKE DNL, BUT WITH GÖDEL T-(CO)NORM AND NOISY WEIGHTS

Substituting Gödel t-(co)norm as chosen fuzzy operations and the Binary Concrete distribution as $g$ into 4 and 5, we get:

$$f_{LFL\_AND}(\boldsymbol{x}^n) = \min_i \left(1 - \min \left(\sigma \left(w_i + \eta \left(\log(u_i) - \log(1 - u_i)\right)\right), 1 - x_i\right)\right) \qquad (14)$$

$$f_{LFL\_OR}(\boldsymbol{x}^n) = \max_i \left(\min \left(\sigma \left(w_i + \eta \left(\log(u_i) - \log(1 - u_i)\right)\right), x_i\right)\right) \qquad (15)$$

where $u_i$ is sampled independently from uniform distribution in $(0, 1)$ and $\eta$ is a hyperparameter controlling the noise scale[3]. An LFL-Type1 network can thus be constructed by arbitrarily combining them and the NOT neurons 6.

### 2.3.2 LFL-TYPE2: LEARNING A LOOK-UP TABLE LIKE DSL, BUT WITH DENSE, NOISY SYMBOLS AND WEIGHTS

DSL used $\epsilon$-greedy policy to produce sparse, noisy symbolic input and predictions in 9 and 13. In LFL-Type2 we replace them with dense, noisy values generated by the Concrete distribution:

$$x_k = \text{softmax}_j \left(N_{ij} - \eta_0 \log \left(-\log \left(u_k\right)\right)\right) \qquad (16)$$

$$m_{lp} = \text{softmax}_p \left(w_{lp} - \eta_1 \log \left(-\log \left(u_{lp}\right)\right)\right) \qquad (17)$$

where $u_k$ and $u_{lp}$ are sampled independently from uniform distrubution in $(0, 1)$, $\eta_0$ and $\eta_1$ are hyperparameters controlling the noise scales. 16 is also used when wrapping LFL-Type1 or LFL-Type3 into NeSy predictors.

### 2.3.3 LFL-TYPE3: USING FROZEN AND LAYER LIKE DSL, BUT WITH INDEPENDENT WEIGHTS FOR THE OR LAYER

In LFL-Type1 all memberships are independently trained, while LFL-Type2 consists of a frozen, predefined layer of AND neurons and a layer of OR neurons where memberships correlate through $\text{softmax}$. Then it's natural to experiment with a third design, in which LFL-Type2's OR layer 17 is replaced by a layer of LFL-Type1's OR neurons where all memberships are freely trainable:

$$m_{lp} = \sigma \left(w_{lp} + \eta_1 \left(\log(u_{lp}) - \log(1 - u_{lp})\right)\right) \qquad (18)$$

### 2.4 NETWORK ARCHITECTURE FOR LFL-BASED DIFFERENTIABLE NESY PREDICTORS

In this subsection, we describe the network architectures that wrap LFL modules into NeSy predictors for direct or recurrent NeSy tasks defined in DSL, such as MNIST Sum and MNIST Multi-digit Sum. How these architectures are applied to the MNIST Arithmetic tasks is shown in Figure 3.

### 2.4.1 DSL'S VANILLA ARCHITECTURE

In DSL, the differentiable logic module is chained after CNN classifiers so that the whole network works as a NeSy predictor. In this work, we add linear reconstruction models that allow us to visualize the symbols learned by CNN classifiers, with stop gradient operations so that they don't affect other network components during training, as shown in Figure 3(a).

For recurrent NeSy tasks, recurrent values (such as $\boldsymbol{c}$) are passed forward through the recurrent blocks, as shown in Figure 3(c).

---

[3]The original paper (Maddison et al., 2016) wrote the Binary Concrete distribution in a slightly different form, where $\log \alpha$ replaced the control parameter $w_i$ and a "temperature" parameter $\lambda$ that divides both the noise and the control parameter replaced $\eta$. The same difference applies to the softmax-like Concrete distribution used in section 2.3.2.

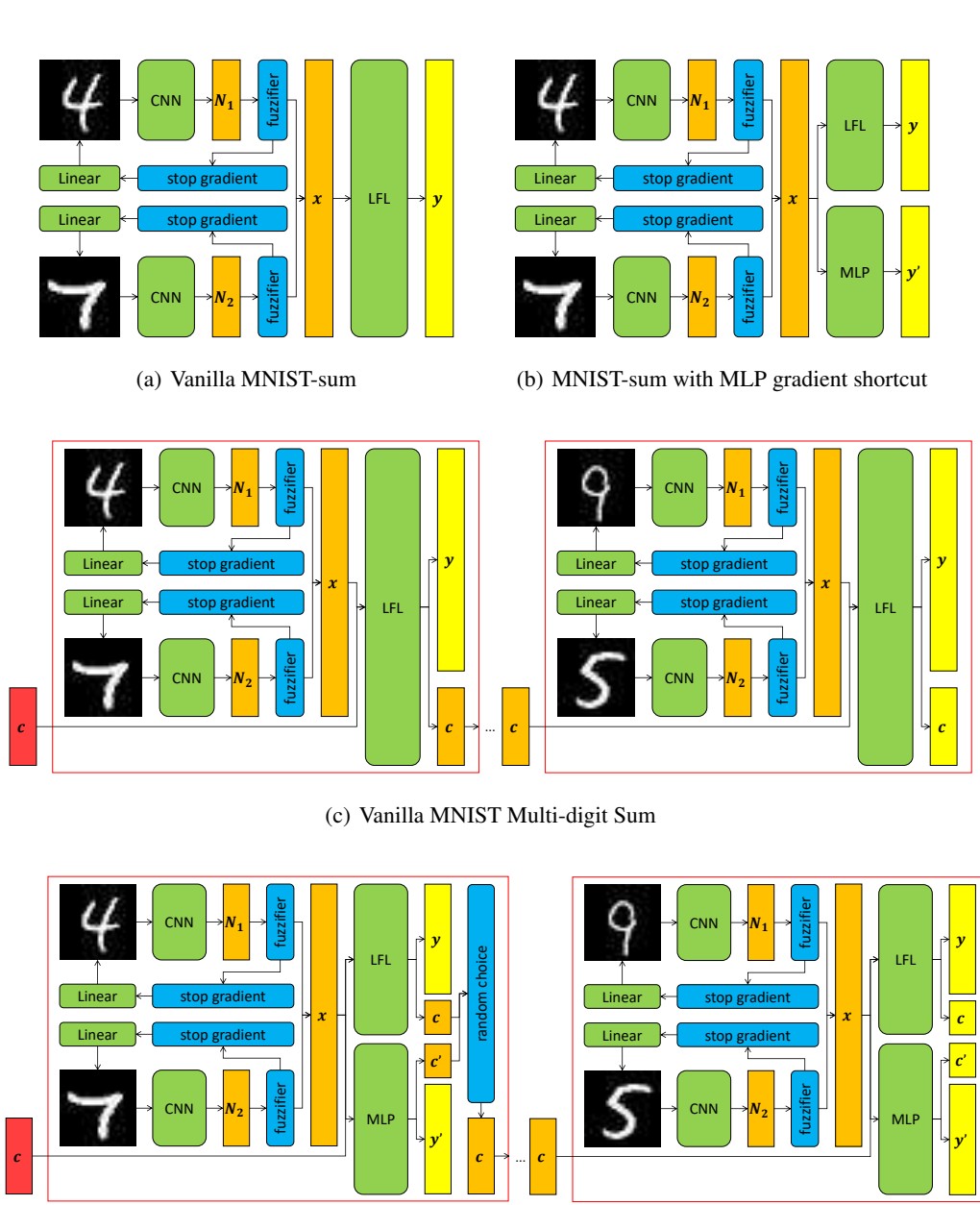

(a) Vanilla MNIST-sum

(b) MNIST-sum with MLP gradient shortcut

(c) Vanilla MNIST Multi-digit Sum

(d) MNIST Multi-digit Sum with MLP gradient shortcut

Figure 3: Network architectures for MNIST Arithmetic tasks, where $N$ means real-valued CNN predictions, "fuzzifier" means softmax-like transformation such as 9 or 16, $x$ and $y$ mean the LFL module's input and output symbols, $c$ means the carry symbols in MNIST Multi-digit Sum, "stop gradient" means detaching the tensor to block gradient backpropagation, "random choice" means randomly choosing one of the input tensors as output for each sample, and each red square contains a recurrent block for MNIST Multi-digit Sum. Parameters are shared among CNNs or linear reconstruction layers. Similar to the membership values in LFL, the fuzzifier layers also output binary, deterministic symbols during inference.

### 2.4.2 OUR NEW TRICK: ADDING AN MLP AS GRADIENT SHORTCUT

In our early experiments, we found that using freely trainable LFL variations like dNL or LFL-Type1 in the vanilla architecture results in a network that struggles to converge. A similar convergence problem has been encountered in training deep CNNs, where it was resolved by adding "shortcut" connections that allow gradients to back-propagate through (He et al., 2016). Inspired by this, we add an MLP that shares the same input and output with the LFL module so that gradients can back-propagate through the MLP, allowing the whole network to converge into an optimal or sub-optimal solution, as shown in Figure 3(b). The MLP uses sigmoid activation in its output layer for consistency.

For recurrent NeSy tasks, the LFL and MLP predict two different versions of the recurrent values $c$, as shown in Figure 3(d). During training, each of them is randomly selected at sample level with probability 0.5 for the next recurrence; during inference, the LFL's prediction is used for the next recurrence.

### 2.5 LOSS FUNCTIONS AND REGULARIZATIONS

For all network architectures described above, the LFL's supervision loss $\mathcal{L}_{sup}$ is measured by the binary cross-entropy loss. For DSL, the loss applies only on its non-zero prediction values.

Another loss term is an L2 reconstruction loss $\mathcal{L}_{rec}$ that trains the linear reconstruction layer. The stop gradient operation stops the reconstruction loss from affecting other modules.

Similar to EQL, we need sparsity constraints to make the LFL modules learn logical formulas in their relatively simple forms. In DSL and LFL-Type2 all trainable memberships are generated with softmax-like transformations 13 17 so that they are always sparse. For memberships generated with sigmoid-like transformations 7 8 14 15 18, we apply a sparsity constraint loss:

$$\mathcal{L}_{reg} = \sum_k \frac{1}{n_k} \sum_{ij} \sigma(w_{ijk}) \tag{19}$$

where $\sigma(w_{ijk})$ is the median membership of the $i$th input of the $j$th neuron of the $k$th layer with sigmoid-like memberships, and $n_k$ means the number of such memberships in the $k$th layer.

For network architectures with the MLP gradient shortcut, another binary cross-entropy loss $\mathcal{L}_{MLP}$ applies on the MLP's prediction. In these networks we also constraint the distribution of LFLs' input symbol $x$ such that the average of $x$ in each batch is close to even distribution over each neural classifier's predicted labels:

$$\mathcal{L}_{label} = \sum_i \frac{1}{n_i} \sum_j \text{BCE}(\frac{1}{n_i}, \frac{1}{b} \sum_k x_{ijk}) \tag{20}$$

where BCE means the binary cross-entropy loss, $x_{ijk}$ means the normalized LFL input corresponding to the $j$th class predicted by the $i$th classifier from the $k$th sample in a batch, b means the batch size, and $n_i$ means classifier $i$'s number of predicted classes.

So the overall loss function used in our experiments is

$$\mathcal{L} = \mathcal{L}_{sup} + \mathcal{L}_{rec} + \beta_1 \mathcal{L}_{reg} + \beta_2 \mathcal{L}_{MLP} + \beta_3 \mathcal{L}_{label} \tag{21}$$

for the network architectures with MLP gradient shortcut. Only $\mathcal{L}_{sup}$, $\mathcal{L}_{rec}$ and $\mathcal{L}_{reg}$ are used for those without the MLP gradient shortcut, and only $\mathcal{L}_{sup}$ and $\mathcal{L}_{rec}$ are used for those with DSL or LFL-Type2. $\beta_1$, $\beta_2$ and $\beta_3$ are hyperparameters adjusting the loss terms' weights.

## 3 EXPERIMENTS

In previous works, dNL has been experimented on ILP tasks that require learning logical formulas from binary training data, and DSL has been experimented on MNIST Arithmetic tasks. We

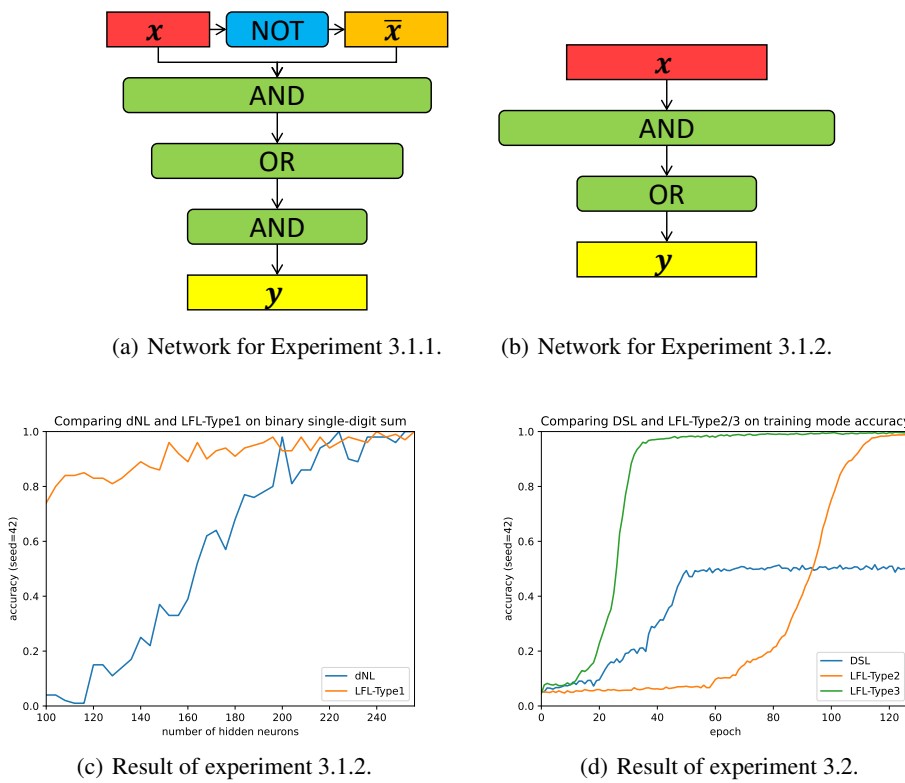

(a) Network for Experiment 3.1.1.  (b) Network for Experiment 3.1.2.

(c) Result of experiment 3.1.2.  (d) Result of experiment 3.2.

Figure 4: Figures related to the experiments.

Table 1: Results of MNIST Arithmetic experiments. **MLP** means whether MLP gradient shortcut is used, $\mathcal{L}_{label}$ means whether $\mathcal{L}_{label}$ is used, **Correct formula** means whether the LFL learns a correct formula. For MNIST Multi-digit Sum two accuracies are evaluated on test datasets with $n_{digit} = 3$ and $n_{digit} = 128$.

| | MLP | $\mathcal{L}_{label}$ | Task | Accuracy(%) | Correct formula |
|---|---|---|---|---|---|
| DSL | ✗ | ✗ | Sum | 97.8 | ✓ |
| LFL-Type2 | ✗ | ✗ | Sum | 98.1 | ✓ |
| LFL-Type3 | ✗ | ✗ | Sum | 98.1 | ✓ |
| | ✗ | ✗ | Multi-digit Sum | 97.7/96.9 | ✓ |
| | ✓ | ✓ | Sum | 98.3 | ✓ |
| | ✗ | ✓ | Sum | 16.3 | ✗ |
| LFL-Type1 | ✓ | ✗ | Sum | 65.8 | ✗ |
| | ✓ | ✓ | Multi-digit Sum | 98.5/98.1 | ✓ |
| | ✗ | ✓ | Multi-digit Sum | 20.2/10.3 | ✗ |
| | ✓ | ✗ | Multi-digit Sum | 81.8/75.0 | ✗ |

evaluate and compare LFL variations on these task types. Test set accuracies of MNIST Arithmetic experiments are shown in Table 1. Note that difference in accuracies among successful MNIST Arithmetic experiments doesn't really matter because mistakes are caused only by CNN classifiers when the LFL learns a correct formula.

## 3.1 COMPARING dNL AND LFL-TYPE1 ON LEARNING LOGICAL FORMULA FROM BINARY DATA

Since dNL has been proven to work well on learning simple logical formulas with 2-layer network architectures (Payani & Fekri, 2019a), we evaluate dNL and LFL-Type1 on two possibly harder tasks: learning a 3-layer logical formula with NOT neurons and learning MNIST Sum's logical formula with limited number of hidden neurons.

### 3.1.1 LEARNING A 3-LAYER LOGICAL FORMULA WITH NEGATION

The 3-layer logical formula to be learned has 8 input concepts and 2 outputs. The 8 input concepts and their negations form 16 input concepts, and the formula then requires at least 8 hidden neurons in an AND layer and 4 neurons in an OR layer to learn (Appendix B). Both dNL and LFL-Type1 learn it successfully with the architecture in Figure 4(a)

### 3.1.2 LEARNING MNIST SUM'S FORMULA WITH LIMITED NUMBER OF HIDDEN NEURONS

The MNIST Sum task requires learning a 2-layer logical formula with 20 inputs, 19 outputs, and 100 hidden concepts. Both dNL and LFL-Type1 can learn it from binary training data with 256 hidden neurons, but LFL-Type1 maintains higher accuracies when the number of hidden neurons reduces towards 100 (Figure 4(c)). The network architecture shown in Figure 4(b) is used for this task and the following experiments where LFL-Type1 is used in NeSy predictors.

## 3.2 COMPARING DSL AND LFL-TYPE2 ON MNIST SUM

LFL-Type2 has the same combinatorial search freedom as DSL's logic module and converges as well as the latter on MNIST Sum. A minor limitation of DSL is that its $\epsilon$-greedy search policy keeps producing random label choices with probability $\epsilon$ after convergence, creating a significant gap between training and inference behaviors. The limitation doesn't exist in LFL-Type2's Concrete noise distribution, as shown in Figure 4(d).

## 3.3 EXPERIMENTING LFL-TYPE3 AND LFL-TYPE1 ON MNIST SUM AND MULTI-DIGIT SUM

LFL-Type3 and LFL-Type1 have larger combinatorial search freedom than DSL, and we evaluate them on MNIST Sum and Multi-digit Sum. LFL-Type3 converges well on MNIST Sum and Multi-digit Sum using DSL's vanilla network architecture, and it actually converges faster than DSL or LFL-Type2 with automatically tuned hyperparameters, as shown in Figure 4(d). LFL-Type1 also converges well with our new architecture that contains the MLP gradient shortcut. In MNIST Sum the formula learned by LFL-Type1 is the same as that of LFL-Type3. In MNIST Multi-digit Sum LFL-Type1 learns a formula different from LFL-Type3 for predicting the next carry, proving LFL-Type1's ability to learn more arbitrary logical formulas in NeSy predictors. See Appendix D for the learned formulas.

## 3.4 ABLATION STUDY

Two additional tricks have been used in NeSy predictors with LFL-Type1: the MLP gradient shortcut and the classification label distribution loss 20. Removing either of them results in failure of joint convergence on MNIST-Sum and MNIST Multi-digit Sum.

# 4 RELATED WORKS

## 4.1 NEURO-SYMBOLIC PREDICTORS

The recent interest in NeSy predictors started with DeepProbLog (Manhaeve et al., 2018), in which programs written in ProbLog (De Raedt et al., 2007) are translated into differentiable arithmetic circuits connected with neural networks, allowing for end-to-end training. ABLSim (Huang et al., 2021) trains neural networks by minimizing the inconsistency between them and given background

knowledge. Logic Tensor Networks (Badreddine et al., 2022) encode symbolic knowledge into differentiable regularizations for training neural networks. DeepStochLog (Winters et al., 2022) extends DeepProbLog with stochastic definite clause grammars. NeurASP (Yang et al., 2023) improves pre-trained neural networks with given Answer Set Programs.

The above methods require given symbolic knowledge, while some other NeSy predictors require iterative training. DeepLogic (Duan et al., 2022) iteratively trains a non-differentiable Deep Logic Module with neural networks. NSIL (Cunnington et al., 2022) iteratively trains an ASP program with neural classifiers. NTOC (Liu et al., 2023) iteratively trains neural networks and symbolic rules, where the symbolic rules can be translated into fixed differentiable circuits. Then DSL (Daniele et al., 2022) is our most closely related work, in which a differentiable logic module that equates a look-up table and fits into the LFL framework is wrapped into a NeSy predictor.

### 4.2 DIFFFERENTIABLE MODULES FOR SYMBOLIC REGRESSION

This work takes initial inspiration from EQL (Martius & Lampert, 2016). An EQL network equates an arithmetic expression after convergence in a way similar to LFL networks, originally proposed for Symbolic Regression. $EQL^{\div}$ (Sahoo et al., 2018) and GMEQL (Chen, 2020) explore different EQL variations, while OccamNet (Dugan et al., 2020) and KAN (Liu et al., 2024) extend the idea with more network designs. EQL's differentiability allows people to connect it with CNNs for joint learning (Kim et al., 2020) and NeSy RL (Luo et al., 2024).

### 4.3 INTEGRAGING NEURAL NETWORKS FOR INDUCTIVE LOGIC PROGRAMMING

As another closely related method, dNL fits into the LFL framework and has been experimented on ILP tasks (Payani & Fekri, 2019a). NLN (Payani & Fekri, 2019b) extends dNL with XOR neurons, and Payani & Fekri (2020) uses dNL to incorporate symbolic knowledge for RL. Works such as $\partial$ILP (Evans & Grefenstette, 2018), NLIL (Yang & Song, 2019), and DLM (Zimmer et al., 2021) also integrate neural networks for ILP tasks. LNN (Riegel et al., 2020; Sen et al., 2022) proposes customized neurons with Łukasiewicz t-(co)norm for ILP tasks, but their neuron designs don't strictly fit into the LFL framework.

## 5 CONCLUTION AND FUTURE WORK

In this work, we present the Logical Formula Learner framework and three novel designs within it. The LFL framework summarizes previous designs into a general framework of network modules that explicitly equate a logical formula after convergence. The proposed LFL-Type1 and LFL-Type2 show improvements over previous designs, and LFL-Type3 works fine with combinatorial search freedom between the two. Furthermore, by wrapping LFL-Type1 into NeSy predictors with MLP gradient shortcut, we obtain the first end-to-end differentiable NeSy predictor that converges from scratch in one single run and explicitly learns an arbitrary logical formula.

Future directions based on this work include:

1. Experimenting with other possible neuron designs within the LFL framework, such as using other t-(co)norm choices or introducing annealing schedules for the noise scales.

2. Using logical neurons defined in this work in layers other than fully connected ones. For example, using them in convolutional layers may produce white-box CNNs with better interpretability or adversarial robustness.

3. Integrating LFL networks into other neural network applications, such as image or video caption, model-free RL, model-based RL, etc. Particularly, integrating LFLs into model-based RL agents may allow such agents to describe their internal representations with sound symbolic logic, opening up a new path toward reliable language generation and understanding.

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

## A    IMPLEMENTATION DETAILS

All experiments are implemented with Python. The neural network modules are implemented with Pytorch, and symbolic expressions are extracted from LFL modules' trained weights with SymPy. MadGrad optimizer (Defazio & Jelassi, 2022) is used for all trainable modules, with learning rates independently tuned for CNN, LFL, and MLP modules. Random seed 42 is used for all experiments, and hyperparameters are selected automatically with Optuna's (Akiba et al., 2019) implementation of Tree-Structured Parzen Estimator (Watanabe, 2023). Pytorch's GPU version is required to run the experiments efficiently.

## B    DATASET DESCRIPTION

Experiment 3.1.1 uses a binary dataset with size $2^8 = 256$. The inputs and outputs are generated with the following logical formula:

$$
\begin{aligned}
\boldsymbol{x} &= [x_0, x_1, \ldots, x_7] \\
\boldsymbol{h}_1 &= [x_0 \wedge x_1, x_2 \wedge x_3, x_4 \wedge x_5, x_6 \wedge x_7, \overline{x_0} \wedge \overline{x_1}, \overline{x_2} \wedge \overline{x_3}, \overline{x_4} \wedge \overline{x_5}, \overline{x_6} \wedge \overline{x_7}] \\
\boldsymbol{h}_2 &= [h_{10} \vee h_{11}, h_{12} \vee h_{13}, h_{14} \vee h_{15}, h_{16} \vee h_{17}] \\
\boldsymbol{y} &= [h_{20} \wedge h_{21}, h_{22} \wedge h_{23}]
\end{aligned}
\tag{22}
$$

where $\boldsymbol{x}$ means the dataset's input, $\boldsymbol{y}$ means its output, $\boldsymbol{h}_1$ and $\boldsymbol{h}_2$ mean intermediate variables that hidden neurons might learn to represent.

Experiment 3.1.2 uses a binary dataset with size $10^2 = 100$. The inputs and outputs are generated with the following logical formula:

$$
\begin{aligned}
\boldsymbol{l} &= [l_0, l_1, \ldots, l_9] \\
\boldsymbol{r} &= [r_0, r_1, \ldots, r_9] \\
\boldsymbol{y} &= [y_0, y_1, \ldots, y_{18}] \\
y_k &= \bigvee_{i+j=k} (l_i \wedge r_j)
\end{aligned}
\tag{23}
$$

where $\boldsymbol{l}$ and $\boldsymbol{r}$ mean input symbols representing values of the two digits, $y_k$ means the output symbol representing that the sum is $k$. This is the same formula that LFL modules are expected to learn when wrapped into a NeSy predictor for MNIST Sum.

MNIST Sum is a standard benchmark for testing NeSy predictors, first used in Manhaeve et al. (2018). Each sample of the MNIST Sum task takes 2 handwritten digits as input and requires a model to predict their sum in $[0, 18]$ as a 19-class classification task. For example, a sample of the dataset has input (4, 7) and classification label 11.

MNIST Multi-digit Sum is an extension of MNIST Sum that requires a model to predict labels for summing two multi-digit handwritten numbers. The two numbers' length $n_{digits}$ is a property of the dataset. The inputs and labels are provided in reverse order. For example, a sample of the dataset with $n_{digits} = 2$ has input [(9, 7),(5, 4)] and classification label [6,0,1] (since 59+47=106).

The MNIST Sum training dataset uses two MNIST images from its training set for each sample, so its actual size is $60000^2 = 3.6 \times 10^9$, making it impractical to actually train a full epoch. In our implementations, we take 8192 randomly generated samples as a pseudo-epoch of the dataset. To generate a sample, we first randomly choose two digits in 0 to 9, then randomly select two handwritten digit images that correspond to the chosen digits.

The MNIST Multi-digit Sum training dataset is implemented in a similar way, where 8192 randomly generated samples are treated as a pseudo-epoch.

The MNIST Sum test dataset uses MNIST test set images instead of training set images and treats 65536 samples as a pseudo-epoch. The MNIST Multi-digit Sum test dataset also uses MNIST test set images, treating 65536 samples as a pseudo-epoch for $n_{digit} = 3$ and 8192 samples as a pseudo-epoch for $n_{digit} = 128$. The accuracies in Table 1 are evaluated on one pseudo-epoch of test datasets.

## C   HYPERPARAMETERS

Since the linear reconstruction layer and its reconstruction loss term don't affect other network components, we fix their loss weight to 1 and learning rate to 0.001. For our novel LFL variations, we describe the manually chosen hyperparameters in text and put the hyperparameters chosen by Optuna in tables. For DSL we use default hyperparameters found in the original implementation.

### C.1   EXPERIMENT 3.1.1

Both dNL and LFL-Type1 are trained for 256 epochs, with 32 and 16 neurons in their hidden layers.

dNL's Hyperparameters:

|  | Meaning | Value |
|---|---|---|
| $lr$ | learning rate | 60.74940936219056 |
| $\beta_1$ | regularization loss weight | 0.13347185900881423 |

LFL-Type1's Hyperparameters:

|  | Meaning | Value |
|---|---|---|
| $lr$ | learning rate | 28.353628505319445 |
| $\eta_1$ | noise scale of the 1st layer | 0.8044418562388825 |
| $\eta_2$ | noise scale of the 2nd layer | 0.09504842110818068 |
| $\eta_3$ | noise scale of the 3rd layer | 0.42150252547988554 |
| $\beta_1$ | regularization loss weight | 0.05891286543711857 |

### C.2   EXPERIMENT 3.1.2

Both dNL and LFL-Type1 are trained for 4096 epochs in every run, allowing them to fully converge with a limited number of hidden neurons. The hyperparameters are chosen with 256 hidden neurons in each network.

dNL's Hyperparameters:

|  | Meaning | Value |
|---|---|---|
| $lr$ | learning rate | 3.721338105780261 |
| $\beta_1$ | regularization loss weight | 0.5163968434483255 |

LFL-Type1's Hyperparameters:

|  | Meaning | Value |
|---|---|---|
| $lr$ | learning rate | 20.743324016859198 |
| $\eta_1$ | noise scale of the 1st layer | 0.9772471411825822 |
| $\eta_2$ | noise scale of the 2nd layer | 0.8185423593180111 |
| $\beta_1$ | regularization loss weight | 0.1485845565420301 |

### C.3   DSL ON MNIST SUM

The network is trained for 128 pseudo-epochs.

|  | Meaning | Value |
|---|---|---|
| $lr_{LFL}$ | the logic module's learning rate | 0.11639833786002995 |
| $lr_{CNN}$ | the CNN's learning rate | 0.001 |
| $\epsilon_{symbol}$ | $\epsilon$ of the fuzzifier | 0.2807344052335263 |
| $\epsilon_{rule}$ | $\epsilon$ of the OR layer | 0.1077119516324264 |

### C.4   LFL-TYPE2 ON MNIST SUM

The network is trained for 128 pseudo-epochs.

| | Meaning | Value |
|---|---|---|
| $lr_{LFL}$ | the LFL's learning rate | 0.03233368589774149 |
| $lr_{CNN}$ | the CNN's learning rate | 0.0013455349127554875 |
| $\eta_0$ | noise scale of the fuzzifier | 0.21398889643701835 |
| $\eta_1$ | noise scale of the OR layer | 0.9837843300240907 |

## C.5   LFL-TYPE3 ON MNIST SUM

The network is trained for 128 pseudo-epochs.

| | Meaning | Value |
|---|---|---|
| $lr_{LFL}$ | the LFL's learning rate | 9.613947918454464 |
| $lr_{CNN}$ | the CNN's learning rate | 0.0019151515603726996 |
| $\eta_0$ | noise scale of the fuzzifier | 0.9113745711049871 |
| $\eta_1$ | noise scale of the OR layer | 0.5785108825130485 |
| $\beta_1$ | regularization loss weight | 0.07854073377893649 |

## C.6   LFL-TYPE3 ON MNIST MULTI-DIGIT SUM

The network is trained for up to 256 pseudo-epochs on datasets with $n_{digits} = 1, 2, 3$ sequentially, with early stopping if the average supervision BCE loss falls below 0.001 at any epoch.

| | Meaning | Value |
|---|---|---|
| $lr_{LFL}$ | the LFL's learning rate | 4.505968658051522 |
| $lr_{CNN}$ | the CNN's learning rate | 0.0047565461690141616 |
| $\eta_0$ | noise scale of the fuzzifier | 0.8632755389984255 |
| $\eta_1$ | noise scale of the OR layer | 0.9467031818141409 |
| $\beta_1$ | regularization loss weight | 0.09120659551465411 |

## C.7   LFL-TYPE1 ON MNIST SUM

The network is trained for 128 pseudo-epochs, with 512 hidden neurons in the LFL-Type1 module.

| | Meaning | Value |
|---|---|---|
| $lr_{LFL}$ | the LFL's learning rate | 3.4946338976075064 |
| $lr_{CNN}$ | the CNN's learning rate | 0.0006208460070100641 |
| $lr_{MLP}$ | the MLP's learning rate | 0.0627384382035572 |
| $\eta_0$ | noise scale of the fuzzifier | 0.3761108683081919 |
| $\eta_1$ | noise scale of the AND layer | 0.9385093053957093 |
| $\eta_2$ | noise scale of the OR layer | 0.5471963095839101 |
| $\beta_1$ | regularization loss weight | 0.4498676093863323 |
| $\beta_2$ | MLP loss weight | 21.454211508035232 |
| $\beta_3$ | label distribution loss weight | 3.7151918234461507 |

## C.8   LFL-TYPE1 ON MNIST MULTI-DIGIT SUM

The network is trained for up to 256 pseudo-epochs on datasets with $n_{digits} = 2, 3$ sequentially (training on $n_{digits} = 1$ would allow LFL-Type1 to completely ignore input carries), with early stopping if the average supervision BCE loss falls below 0.001 at any epoch. There are 8192 hidden neurons in the LFL-Type1 module.

| | Meaning | Value |
|---|---|---|
| $lr_{LFL}$ | the LFL's learning rate | 11.331405298874264 |
| $lr_{CNN}$ | the CNN's learning rate | 0.000571482276406197 |
| $lr_{MLP}$ | the MLP's learning rate | 0.012437761948519307 |
| $\eta_0$ | noise scale of the fuzzifier | 0.3115310029581702 |
| $\eta_1$ | noise scale of the AND layer | 0.8485090728796351 |
| $\eta_2$ | noise scale of the OR layer | 0.9770926024145439 |
| $\beta_1$ | regularization loss weight | 0.13687607266487536 |
| $\beta_2$ | MLP loss weight | 8.473281653324152 |
| $\beta_3$ | label distribution loss weight | 1.266068955037883 |

# D    DETAILED EXPERIMENT RESULTS

Here we demonstrate what the NeSy predictors have learned with confusion matrices of their neural classifiers and logical formulas learned by their LFL modules. In these logical formulas we represent digit symbols with their linear reconstruction layer's reconstruction of corresponding one-hot tensors. The reconstructed images are colored in yellow or cyan, representing symbols predicted from two input images. In MNIST Multi-digit Sum we denote the carry symbols as $C_0$ for input carry being 0, $C_1$ for input carry being 1, $C'_0$ for output carry being 0 and $C'_1$ for output carry being 1.

## D.1    DSL ON MNIST SUM

Confusion matrices on MNIST training and test dataset:

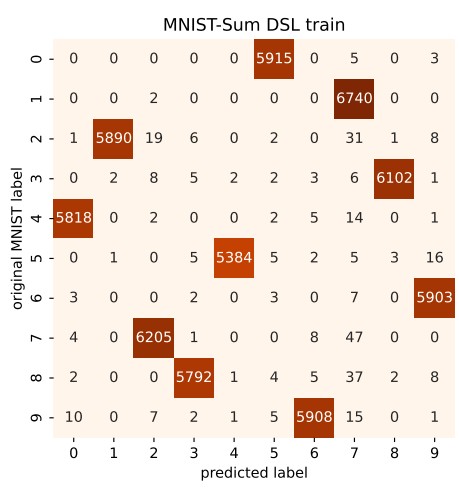 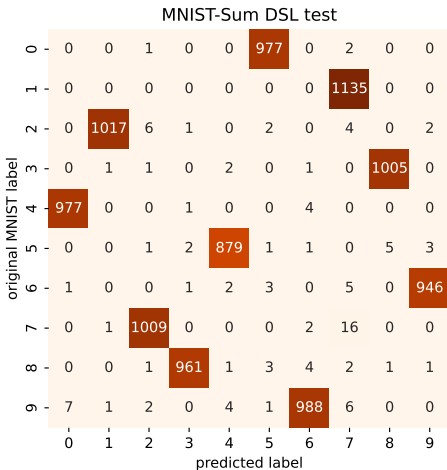

Learned formula:

$y_0 = \boxed{0} \wedge \boxed{0}$

$y_1 = (\boxed{0} \wedge \boxed{1}) \vee (\boxed{1} \wedge \boxed{0})$

$y_2 = (\boxed{2} \wedge \boxed{0}) \vee (\boxed{0} \wedge \boxed{2}) \vee (\boxed{1} \wedge \boxed{1})$

$y_3 = (\boxed{2} \wedge \boxed{1}) \vee (\boxed{0} \wedge \boxed{3}) \vee (\boxed{1} \wedge \boxed{2}) \vee (\boxed{3} \wedge \boxed{0})$

$y_4 = (\boxed{4} \wedge \boxed{0}) \vee (\boxed{2} \wedge \boxed{2}) \vee (\boxed{0} \wedge \boxed{4}) \vee (\boxed{1} \wedge \boxed{3}) \vee (\boxed{3} \wedge \boxed{1})$

$y_5 = (\boxed{4} \wedge \boxed{1}) \vee (\boxed{2} \wedge \boxed{3}) \vee (\boxed{5} \wedge \boxed{0}) \vee (\boxed{0} \wedge \boxed{5}) \vee (\boxed{1} \wedge \boxed{4}) \vee (\boxed{3} \wedge \boxed{2})$

$y_6 = (\boxed{4} \wedge \boxed{2}) \vee (\boxed{2} \wedge \boxed{4}) \vee (\boxed{5} \wedge \boxed{1}) \vee (\boxed{0} \wedge \boxed{6}) \vee (\boxed{1} \wedge \boxed{5}) \vee (\boxed{3} \wedge \boxed{3}) \vee (\boxed{6} \wedge \boxed{0})$

$y_7 = (\boxed{4} \wedge \boxed{3}) \vee (\boxed{2} \wedge \boxed{5}) \vee (\boxed{7} \wedge \boxed{0}) \vee (\boxed{5} \wedge \boxed{2}) \vee (\boxed{0} \wedge \boxed{7}) \vee (\boxed{1} \wedge \boxed{6}) \vee (\boxed{3} \wedge \boxed{4}) \vee (\boxed{6} \wedge \boxed{1})$

$y_8 = (\boxed{4} \wedge \boxed{4}) \vee (\boxed{2} \wedge \boxed{6}) \vee (\boxed{7} \wedge \boxed{1}) \vee (\boxed{8} \wedge \boxed{0}) \vee (\boxed{5} \wedge \boxed{3}) \vee (\boxed{0} \wedge \boxed{8}) \vee (\boxed{1} \wedge \boxed{7}) \vee (\boxed{3} \wedge \boxed{5}) \vee (\boxed{6} \wedge \boxed{2})$

$y_9 = (4 \wedge 5) \vee (2 \wedge 7) \vee (7 \wedge 2) \vee (8 \wedge 1) \vee (5 \wedge 4) \vee (0 \wedge 9) \vee (9 \wedge 0) \vee (1 \wedge 8) \vee (3 \wedge 6) \vee (6 \wedge 3)$

$y_{10} = (4 \wedge 6) \vee (2 \wedge 8) \vee (7 \wedge 3) \vee (8 \wedge 2) \vee (5 \wedge 5) \vee (9 \wedge 1) \vee (1 \wedge 9) \vee (3 \wedge 7) \vee (6 \wedge 4)$

$y_{11} = (4 \wedge 7) \vee (2 \wedge 9) \vee (7 \wedge 4) \vee (8 \wedge 3) \vee (5 \wedge 6) \vee (9 \wedge 2) \vee (3 \wedge 8) \vee (6 \wedge 5)$

$y_{12} = (4 \wedge 8) \vee (7 \wedge 5) \vee (8 \wedge 4) \vee (5 \wedge 7) \vee (9 \wedge 3) \vee (3 \wedge 9) \vee (6 \wedge 6)$

$y_{13} = (4 \wedge 9) \vee (7 \wedge 6) \vee (8 \wedge 5) \vee (5 \wedge 8) \vee (9 \wedge 4) \vee (6 \wedge 7)$

$y_{14} = (7 \wedge 7) \vee (8 \wedge 6) \vee (5 \wedge 9) \vee (9 \wedge 5) \vee (6 \wedge 8)$

$y_{15} = (7 \wedge 8) \vee (8 \wedge 7) \vee (9 \wedge 6) \vee (6 \wedge 9)$

$y_{16} = (7 \wedge 9) \vee (8 \wedge 8) \vee (9 \wedge 7)$

$y_{17} = (8 \wedge 9) \vee (9 \wedge 8)$

$y_{18} = 9 \wedge 9$

### D.2 LFL-Type2 on MNIST Sum

Confusion matrices on MNIST training and test dataset:

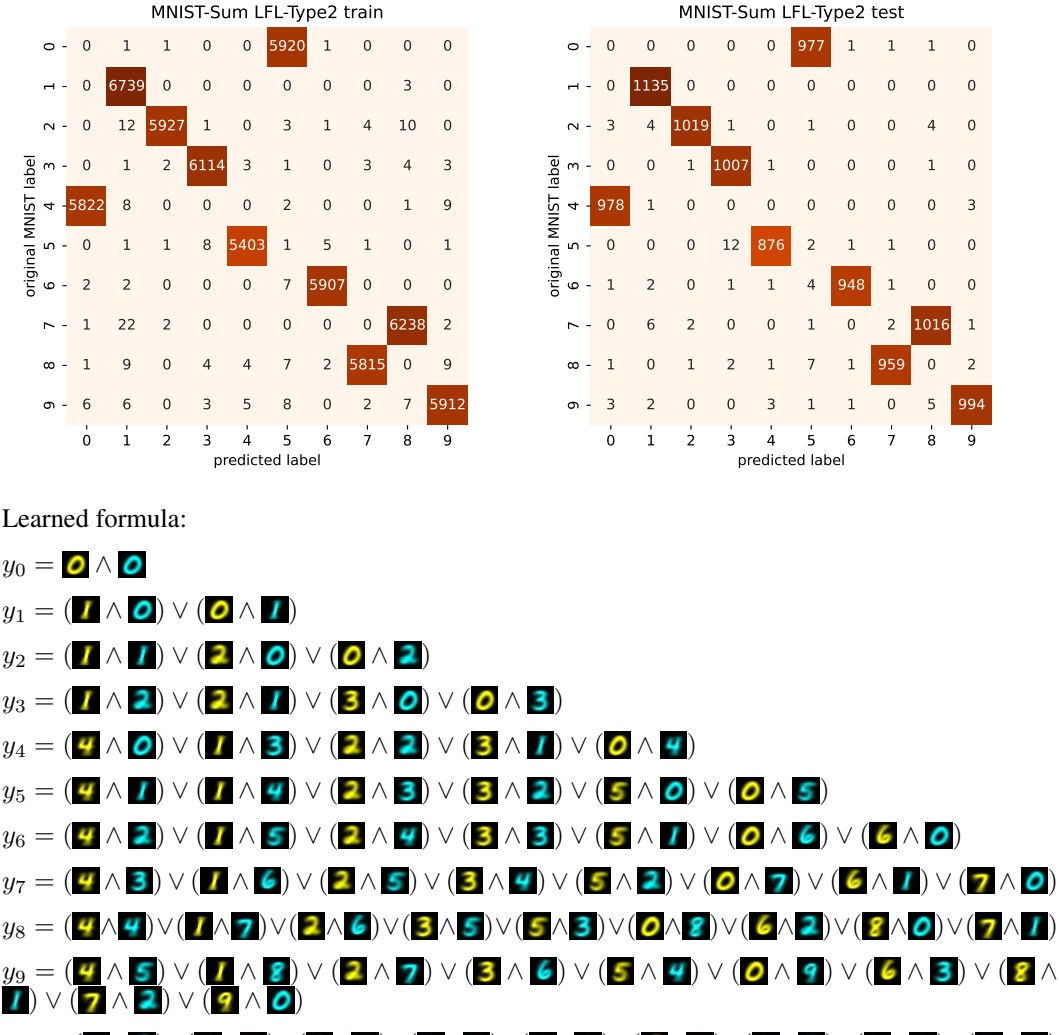

Learned formula:

$y_0 = 0 \wedge 0$

$y_1 = (1 \wedge 0) \vee (0 \wedge 1)$

$y_2 = (1 \wedge 1) \vee (2 \wedge 0) \vee (0 \wedge 2)$

$y_3 = (1 \wedge 2) \vee (2 \wedge 1) \vee (3 \wedge 0) \vee (0 \wedge 3)$

$y_4 = (4 \wedge 0) \vee (1 \wedge 3) \vee (2 \wedge 2) \vee (3 \wedge 1) \vee (0 \wedge 4)$

$y_5 = (4 \wedge 1) \vee (1 \wedge 4) \vee (2 \wedge 3) \vee (3 \wedge 2) \vee (5 \wedge 0) \vee (0 \wedge 5)$

$y_6 = (4 \wedge 2) \vee (1 \wedge 5) \vee (2 \wedge 4) \vee (3 \wedge 3) \vee (5 \wedge 1) \vee (0 \wedge 6) \vee (6 \wedge 0)$

$y_7 = (4 \wedge 3) \vee (1 \wedge 6) \vee (2 \wedge 5) \vee (3 \wedge 4) \vee (5 \wedge 2) \vee (0 \wedge 7) \vee (6 \wedge 1) \vee (7 \wedge 0)$

$y_8 = (4 \wedge 4) \vee (1 \wedge 7) \vee (2 \wedge 6) \vee (3 \wedge 5) \vee (5 \wedge 3) \vee (0 \wedge 8) \vee (6 \wedge 2) \vee (8 \wedge 0) \vee (7 \wedge 1)$

$y_9 = (4 \wedge 5) \vee (1 \wedge 8) \vee (2 \wedge 7) \vee (3 \wedge 6) \vee (5 \wedge 4) \vee (0 \wedge 9) \vee (6 \wedge 3) \vee (8 \wedge 1) \vee (7 \wedge 2) \vee (9 \wedge 0)$

$y_{10} = (4 \wedge 6) \vee (1 \wedge 9) \vee (2 \wedge 8) \vee (3 \wedge 7) \vee (5 \wedge 5) \vee (6 \wedge 4) \vee (8 \wedge 2) \vee (7 \wedge 3) \vee (9 \wedge 1)$

$y_{11} = (4 \wedge 7) \vee (2 \wedge 9) \vee (3 \wedge 8) \vee (5 \wedge 6) \vee (6 \wedge 5) \vee (8 \wedge 3) \vee (7 \wedge 4) \vee (9 \wedge 2)$

$y_{12} = (4 \wedge 8) \vee (3 \wedge 9) \vee (5 \wedge 7) \vee (6 \wedge 6) \vee (8 \wedge 4) \vee (7 \wedge 5) \vee (9 \wedge 3)$

$y_{13} = (4 \wedge 9) \vee (5 \wedge 8) \vee (6 \wedge 7) \vee (8 \wedge 5) \vee (7 \wedge 6) \vee (9 \wedge 4)$

$y_{14} = (5 \wedge 9) \vee (6 \wedge 8) \vee (8 \wedge 6) \vee (7 \wedge 7) \vee (9 \wedge 5)$

$y_{15} = (6 \wedge 9) \vee (8 \wedge 7) \vee (7 \wedge 8) \vee (9 \wedge 6)$

$y_{16} = (8 \wedge 8) \vee (7 \wedge 9) \vee (9 \wedge 7)$

$y_{17} = (8 \wedge 9) \vee (9 \wedge 8)$

$y_{18} = 9 \wedge 9$

## D.3 LFL-Type3 on MNIST Sum

Confusion matrices on MNIST training and test dataset:

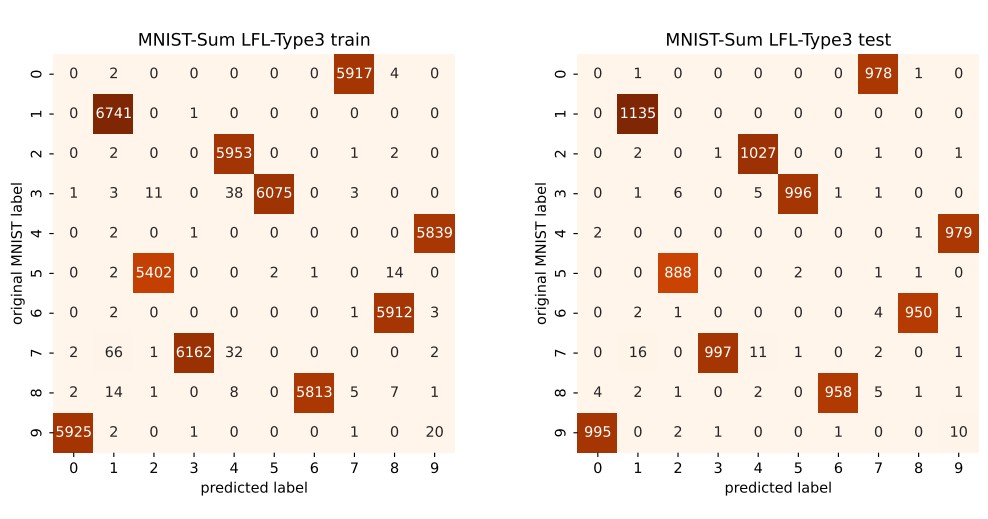

Learned formula:

$y_0 = 0 \wedge 0$

$y_1 = (1 \wedge 0) \vee (0 \wedge 1)$

$y_2 = (1 \wedge 1) \vee (2 \wedge 0) \vee (0 \wedge 2)$

$y_3 = (1 \wedge 2) \vee (2 \wedge 1) \vee (3 \wedge 0) \vee (0 \wedge 3)$

$y_4 = (1 \wedge 3) \vee (2 \wedge 2) \vee (3 \wedge 1) \vee (0 \wedge 4) \vee (4 \wedge 0)$

$y_5 = (1 \wedge 4) \vee (5 \wedge 0) \vee (2 \wedge 3) \vee (3 \wedge 2) \vee (0 \wedge 5) \vee (4 \wedge 1)$

$y_6 = (1 \wedge 5) \vee (5 \wedge 1) \vee (2 \wedge 4) \vee (3 \wedge 3) \vee (0 \wedge 6) \vee (6 \wedge 0) \vee (4 \wedge 2)$

$y_7 = (1 \wedge 6) \vee (5 \wedge 2) \vee (7 \wedge 0) \vee (2 \wedge 5) \vee (3 \wedge 4) \vee (0 \wedge 7) \vee (6 \wedge 1) \vee (4 \wedge 3)$

$y_8 = (1 \wedge 7) \vee (5 \wedge 3) \vee (7 \wedge 1) \vee (2 \wedge 6) \vee (3 \wedge 5) \vee (8 \wedge 0) \vee (0 \wedge 8) \vee (6 \wedge 2) \vee (4 \wedge 4)$

$y_9 = (9 \wedge 0) \vee (1 \wedge 8) \vee (5 \wedge 4) \vee (7 \wedge 2) \vee (2 \wedge 7) \vee (3 \wedge 6) \vee (8 \wedge 1) \vee (0 \wedge 9) \vee (6 \wedge 3) \vee (4 \wedge 5)$

$y_{10} = (9 \wedge 1) \vee (1 \wedge 9) \vee (5 \wedge 5) \vee (7 \wedge 3) \vee (2 \wedge 8) \vee (3 \wedge 7) \vee (8 \wedge 2) \vee (6 \wedge 4) \vee (4 \wedge 6)$

$y_{11} = (9 \wedge 2) \vee (5 \wedge 6) \vee (7 \wedge 4) \vee (2 \wedge 9) \vee (3 \wedge 8) \vee (8 \wedge 3) \vee (6 \wedge 5) \vee (4 \wedge 7)$

$y_{12} = (9 \wedge 3) \vee (5 \wedge 7) \vee (7 \wedge 5) \vee (3 \wedge 9) \vee (8 \wedge 4) \vee (6 \wedge 6) \vee (4 \wedge 8)$

$y_{13} = (9 \wedge 4) \vee (5 \wedge 8) \vee (7 \wedge 6) \vee (8 \wedge 5) \vee (6 \wedge 7) \vee (4 \wedge 9)$

$y_{14} = (9 \wedge 5) \vee (5 \wedge 9) \vee (7 \wedge 7) \vee (8 \wedge 6) \vee (6 \wedge 8)$

$y_{15} = (9 \wedge 6) \vee (7 \wedge 8) \vee (8 \wedge 7) \vee (6 \wedge 9)$

$y_{16} = (9 \wedge 7) \vee (7 \wedge 9) \vee (8 \wedge 8)$

$y_{17} = (9 \wedge 8) \vee (8 \wedge 9)$

$y_{18} = 9 \wedge 9$

## D.4 LFL-TYPE3 ON MNIST MULTI-DIGIT SUM

Confusion matrices on MNIST training and test dataset:

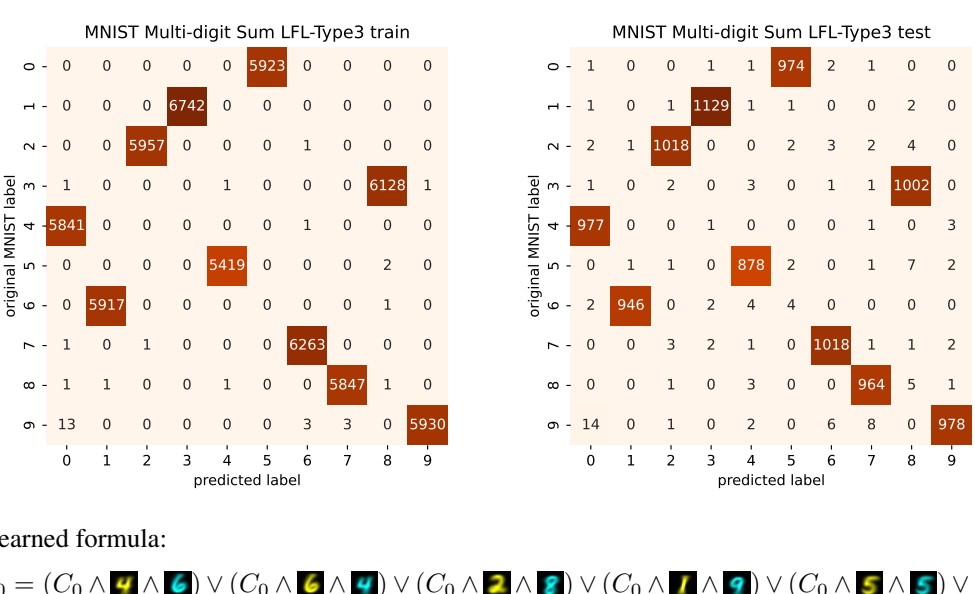

Learned formula:

$y_0 = (C_0 \wedge 4 \wedge 6) \vee (C_0 \wedge 6 \wedge 4) \vee (C_0 \wedge 2 \wedge 8) \vee (C_0 \wedge 1 \wedge 9) \vee (C_0 \wedge 5 \wedge 5) \vee (C_0 \wedge 0 \wedge 0) \vee (C_0 \wedge 7 \wedge 3) \vee (C_0 \wedge 8 \wedge 2) \vee (C_0 \wedge 3 \wedge 7) \vee (C_0 \wedge 9 \wedge 1) \vee (C_1 \wedge 4 \wedge 5) \vee (C_1 \wedge 6 \wedge 3) \vee (C_1 \wedge 2 \wedge 7) \vee (C_1 \wedge 1 \wedge 8) \vee (C_1 \wedge 5 \wedge 4) \vee (C_1 \wedge 0 \wedge 9) \vee (C_1 \wedge 7 \wedge 2) \vee (C_1 \wedge 8 \wedge 1) \vee (C_1 \wedge 3 \wedge 6) \vee (C_1 \wedge 9 \wedge 0)$

$y_1 = (C_0 \wedge 4 \wedge 7) \vee (C_0 \wedge 6 \wedge 5) \vee (C_0 \wedge 2 \wedge 9) \vee (C_0 \wedge 1 \wedge 0) \vee (C_0 \wedge 5 \wedge 6) \vee (C_0 \wedge 0 \wedge 1) \vee (C_0 \wedge 7 \wedge 4) \vee (C_0 \wedge 8 \wedge 3) \vee (C_0 \wedge 3 \wedge 8) \vee (C_0 \wedge 9 \wedge 2) \vee (C_1 \wedge 4 \wedge 6) \vee (C_1 \wedge 6 \wedge 4) \vee (C_1 \wedge 2 \wedge 8) \vee (C_1 \wedge 1 \wedge 9) \vee (C_1 \wedge 5 \wedge 5) \vee (C_1 \wedge 0 \wedge 0) \vee (C_1 \wedge 7 \wedge 3) \vee (C_1 \wedge 8 \wedge 2) \vee (C_1 \wedge 3 \wedge 7) \vee (C_1 \wedge 9 \wedge 1)$

$y_2 = (C_0 \wedge 4 \wedge 8) \vee (C_0 \wedge 6 \wedge 6) \vee (C_0 \wedge 2 \wedge 0) \vee (C_0 \wedge 1 \wedge 1) \vee (C_0 \wedge 5 \wedge 7) \vee (C_0 \wedge 0 \wedge 2) \vee (C_0 \wedge 7 \wedge 5) \vee (C_0 \wedge 8 \wedge 4) \vee (C_0 \wedge 3 \wedge 9) \vee (C_0 \wedge 9 \wedge 3) \vee (C_1 \wedge 4 \wedge 7) \vee (C_1 \wedge 6 \wedge 5) \vee (C_1 \wedge 2 \wedge 9) \vee (C_1 \wedge 1 \wedge 0) \vee (C_1 \wedge 5 \wedge 6) \vee (C_1 \wedge 0 \wedge 1) \vee (C_1 \wedge 7 \wedge 4) \vee (C_1 \wedge 8 \wedge 3) \vee (C_1 \wedge 3 \wedge 8) \vee (C_1 \wedge 9 \wedge 2)$

$y_3 = (C_0 \wedge 4 \wedge 9) \vee (C_0 \wedge 6 \wedge 7) \vee (C_0 \wedge 2 \wedge 1) \vee (C_0 \wedge 1 \wedge 2) \vee (C_0 \wedge 5 \wedge 8) \vee (C_0 \wedge 0 \wedge 3) \vee (C_0 \wedge 7 \wedge 6) \vee (C_0 \wedge 8 \wedge 5) \vee (C_0 \wedge 3 \wedge 0) \vee (C_0 \wedge 9 \wedge 4) \vee (C_1 \wedge 4 \wedge 8) \vee (C_1 \wedge 6 \wedge 6) \vee (C_1 \wedge 2 \wedge 0) \vee (C_1 \wedge 1 \wedge 1) \vee (C_1 \wedge 5 \wedge 7) \vee (C_1 \wedge 0 \wedge 2) \vee (C_1 \wedge 7 \wedge 5) \vee (C_1 \wedge 8 \wedge 4) \vee (C_1 \wedge 3 \wedge 9) \vee (C_1 \wedge 9 \wedge 3)$

$y_4 = (C_0 \wedge 4 \wedge 0) \vee (C_0 \wedge 6 \wedge 8) \vee (C_0 \wedge 2 \wedge 2) \vee (C_0 \wedge 1 \wedge 3) \vee (C_0 \wedge 5 \wedge 9) \vee (C_0 \wedge 0 \wedge 4) \vee (C_0 \wedge 7 \wedge 7) \vee (C_0 \wedge 8 \wedge 6) \vee (C_0 \wedge 3 \wedge 1) \vee (C_0 \wedge 9 \wedge 5) \vee (C_1 \wedge 4 \wedge 9) \vee (C_1 \wedge 6 \wedge 7) \vee (C_1 \wedge 2 \wedge 1) \vee (C_1 \wedge 1 \wedge 2) \vee (C_1 \wedge 5 \wedge 8) \vee (C_1 \wedge 0 \wedge 3) \vee (C_1 \wedge 7 \wedge 6) \vee (C_1 \wedge 8 \wedge 5) \vee (C_1 \wedge 3 \wedge 0) \vee (C_1 \wedge 9 \wedge 4)$

$y_5 = (C_0 \wedge 4 \wedge 1) \vee (C_0 \wedge 6 \wedge 9) \vee (C_0 \wedge 2 \wedge 3) \vee (C_0 \wedge 1 \wedge 4) \vee (C_0 \wedge 5 \wedge 0) \vee (C_0 \wedge 0 \wedge 5) \vee (C_0 \wedge 7 \wedge 8) \vee (C_0 \wedge 8 \wedge 7) \vee (C_0 \wedge 3 \wedge 2) \vee (C_0 \wedge 9 \wedge 6) \vee (C_1 \wedge 4 \wedge 0) \vee (C_1 \wedge 6 \wedge 8) \vee (C_1 \wedge 2 \wedge 2) \vee (C_1 \wedge 1 \wedge 3) \vee (C_1 \wedge 5 \wedge 9) \vee (C_1 \wedge 0 \wedge 4) \vee (C_1 \wedge 7 \wedge 7) \vee (C_1 \wedge 8 \wedge 6) \vee (C_1 \wedge 3 \wedge 1) \vee (C_1 \wedge 9 \wedge 5)$

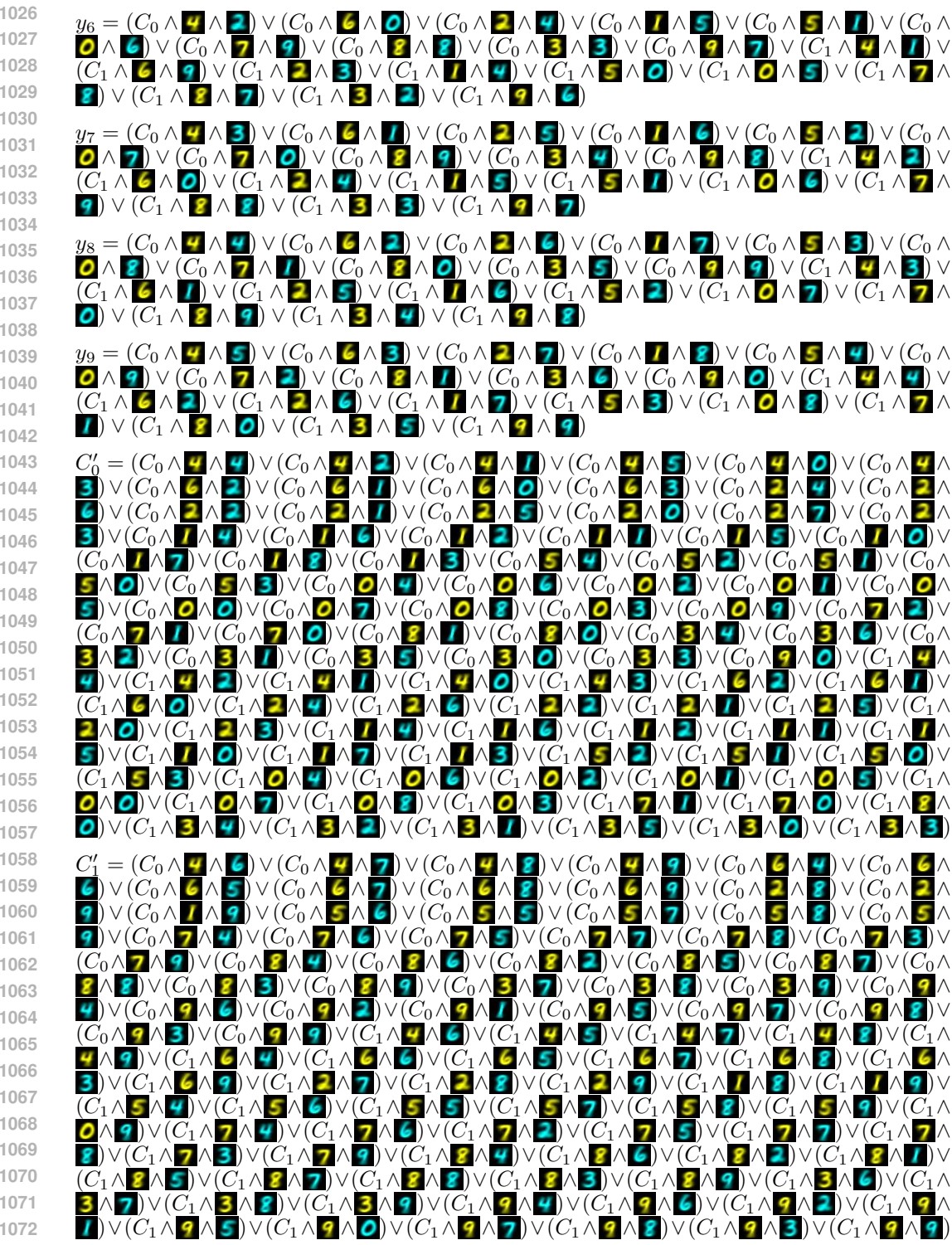

## D.5 LFL-TYPE1 ON MNIST SUM

Confusion matrices on MNIST training and test dataset:

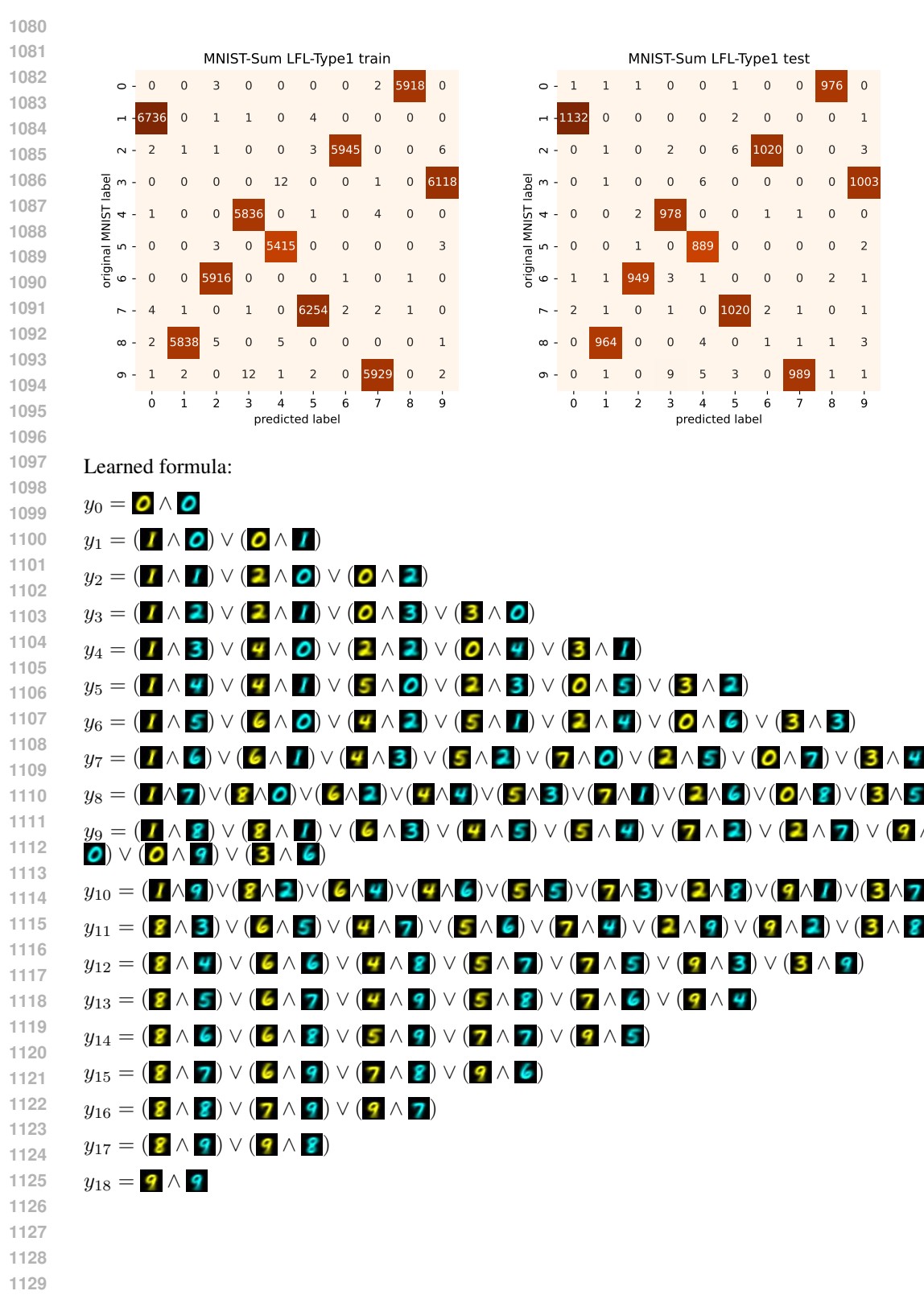

Learned formula:

$y_0 = \boxed{0} \wedge \boxed{0}$

$y_1 = (\boxed{1} \wedge \boxed{0}) \vee (\boxed{0} \wedge \boxed{1})$

$y_2 = (\boxed{1} \wedge \boxed{1}) \vee (\boxed{2} \wedge \boxed{0}) \vee (\boxed{0} \wedge \boxed{2})$

$y_3 = (\boxed{1} \wedge \boxed{2}) \vee (\boxed{2} \wedge \boxed{1}) \vee (\boxed{0} \wedge \boxed{3}) \vee (\boxed{3} \wedge \boxed{0})$

$y_4 = (\boxed{1} \wedge \boxed{3}) \vee (\boxed{4} \wedge \boxed{0}) \vee (\boxed{2} \wedge \boxed{2}) \vee (\boxed{0} \wedge \boxed{4}) \vee (\boxed{3} \wedge \boxed{1})$

$y_5 = (\boxed{1} \wedge \boxed{4}) \vee (\boxed{4} \wedge \boxed{1}) \vee (\boxed{5} \wedge \boxed{0}) \vee (\boxed{2} \wedge \boxed{3}) \vee (\boxed{0} \wedge \boxed{5}) \vee (\boxed{3} \wedge \boxed{2})$

$y_6 = (\boxed{1} \wedge \boxed{5}) \vee (\boxed{6} \wedge \boxed{0}) \vee (\boxed{4} \wedge \boxed{2}) \vee (\boxed{5} \wedge \boxed{1}) \vee (\boxed{2} \wedge \boxed{4}) \vee (\boxed{0} \wedge \boxed{6}) \vee (\boxed{3} \wedge \boxed{3})$

$y_7 = (\boxed{1} \wedge \boxed{6}) \vee (\boxed{6} \wedge \boxed{1}) \vee (\boxed{4} \wedge \boxed{3}) \vee (\boxed{5} \wedge \boxed{2}) \vee (\boxed{7} \wedge \boxed{0}) \vee (\boxed{2} \wedge \boxed{5}) \vee (\boxed{0} \wedge \boxed{7}) \vee (\boxed{3} \wedge \boxed{4})$

$y_8 = (\boxed{1} \wedge \boxed{7}) \vee (\boxed{8} \wedge \boxed{0}) \vee (\boxed{6} \wedge \boxed{2}) \vee (\boxed{4} \wedge \boxed{4}) \vee (\boxed{5} \wedge \boxed{3}) \vee (\boxed{7} \wedge \boxed{1}) \vee (\boxed{2} \wedge \boxed{6}) \vee (\boxed{0} \wedge \boxed{8}) \vee (\boxed{3} \wedge \boxed{5})$

$y_9 = (\boxed{1} \wedge \boxed{8}) \vee (\boxed{8} \wedge \boxed{1}) \vee (\boxed{6} \wedge \boxed{3}) \vee (\boxed{4} \wedge \boxed{5}) \vee (\boxed{5} \wedge \boxed{4}) \vee (\boxed{7} \wedge \boxed{2}) \vee (\boxed{2} \wedge \boxed{7}) \vee (\boxed{9} \wedge \boxed{0}) \vee (\boxed{0} \wedge \boxed{9}) \vee (\boxed{3} \wedge \boxed{6})$

$y_{10} = (\boxed{1} \wedge \boxed{9}) \vee (\boxed{8} \wedge \boxed{2}) \vee (\boxed{6} \wedge \boxed{4}) \vee (\boxed{4} \wedge \boxed{6}) \vee (\boxed{5} \wedge \boxed{5}) \vee (\boxed{7} \wedge \boxed{3}) \vee (\boxed{2} \wedge \boxed{8}) \vee (\boxed{9} \wedge \boxed{1}) \vee (\boxed{3} \wedge \boxed{7})$

$y_{11} = (\boxed{8} \wedge \boxed{3}) \vee (\boxed{6} \wedge \boxed{5}) \vee (\boxed{4} \wedge \boxed{7}) \vee (\boxed{5} \wedge \boxed{6}) \vee (\boxed{7} \wedge \boxed{4}) \vee (\boxed{2} \wedge \boxed{9}) \vee (\boxed{9} \wedge \boxed{2}) \vee (\boxed{3} \wedge \boxed{8})$

$y_{12} = (\boxed{8} \wedge \boxed{4}) \vee (\boxed{6} \wedge \boxed{6}) \vee (\boxed{4} \wedge \boxed{8}) \vee (\boxed{5} \wedge \boxed{7}) \vee (\boxed{7} \wedge \boxed{5}) \vee (\boxed{9} \wedge \boxed{3}) \vee (\boxed{3} \wedge \boxed{9})$

$y_{13} = (\boxed{8} \wedge \boxed{5}) \vee (\boxed{6} \wedge \boxed{7}) \vee (\boxed{4} \wedge \boxed{9}) \vee (\boxed{5} \wedge \boxed{8}) \vee (\boxed{7} \wedge \boxed{6}) \vee (\boxed{9} \wedge \boxed{4})$

$y_{14} = (\boxed{8} \wedge \boxed{6}) \vee (\boxed{6} \wedge \boxed{8}) \vee (\boxed{5} \wedge \boxed{9}) \vee (\boxed{7} \wedge \boxed{7}) \vee (\boxed{9} \wedge \boxed{5})$

$y_{15} = (\boxed{8} \wedge \boxed{7}) \vee (\boxed{6} \wedge \boxed{9}) \vee (\boxed{7} \wedge \boxed{8}) \vee (\boxed{9} \wedge \boxed{6})$

$y_{16} = (\boxed{8} \wedge \boxed{8}) \vee (\boxed{7} \wedge \boxed{9}) \vee (\boxed{9} \wedge \boxed{7})$

$y_{17} = (\boxed{8} \wedge \boxed{9}) \vee (\boxed{9} \wedge \boxed{8})$

$y_{18} = \boxed{9} \wedge \boxed{9}$

## D.6 LFL-Type1 on MNIST Multi-digit Sum

Confusion matrices on MNIST training and test dataset:

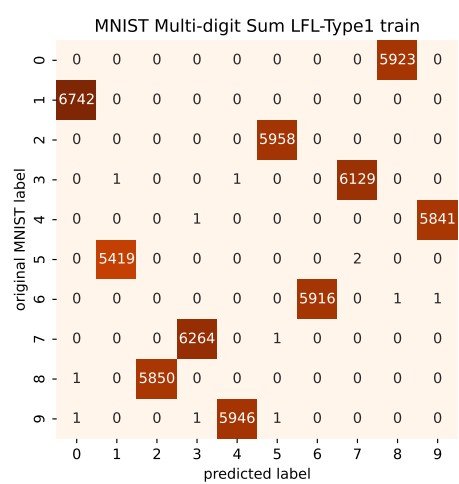

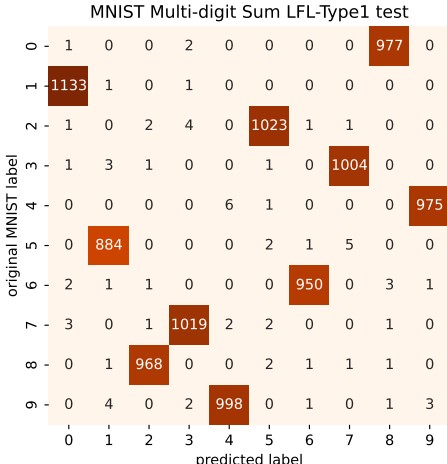

Learned formula:

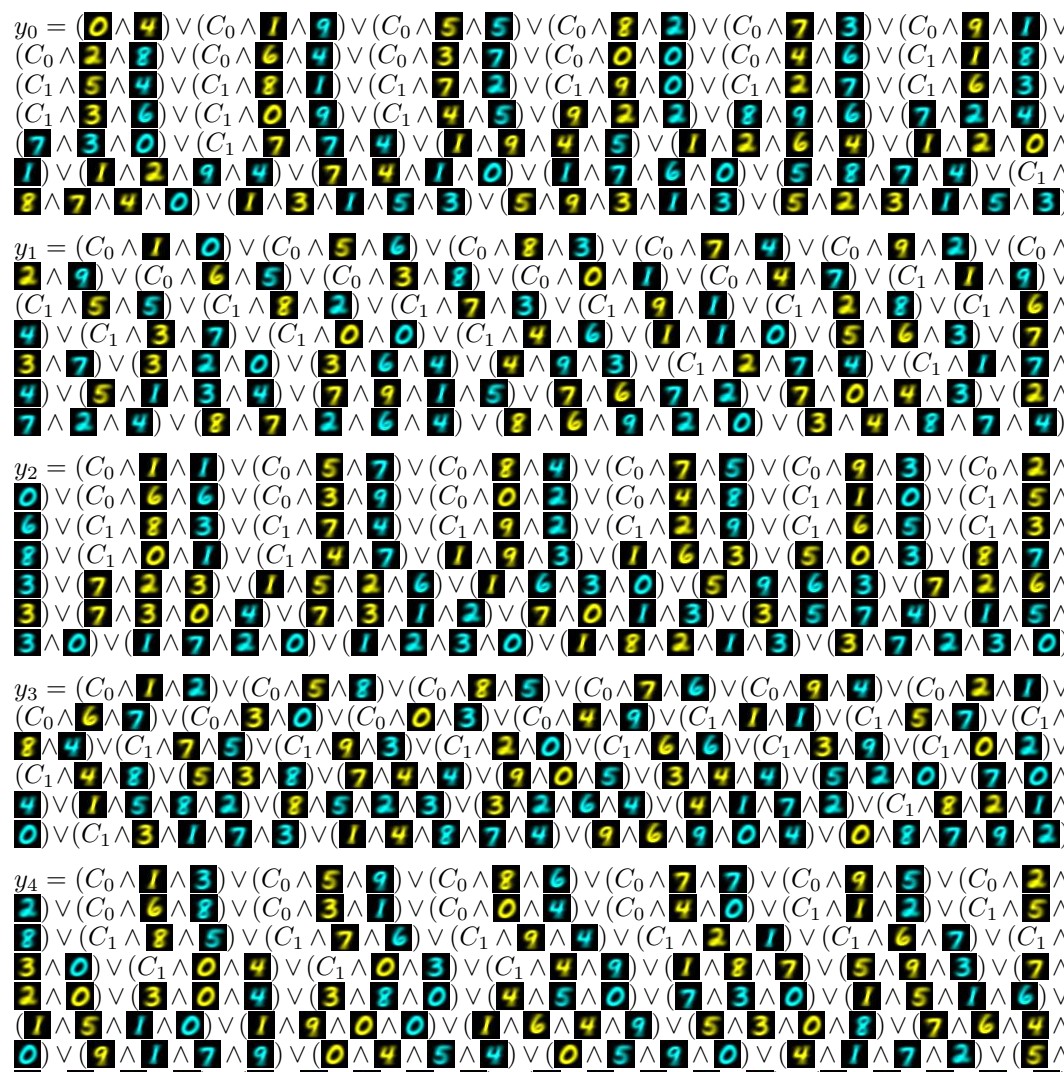

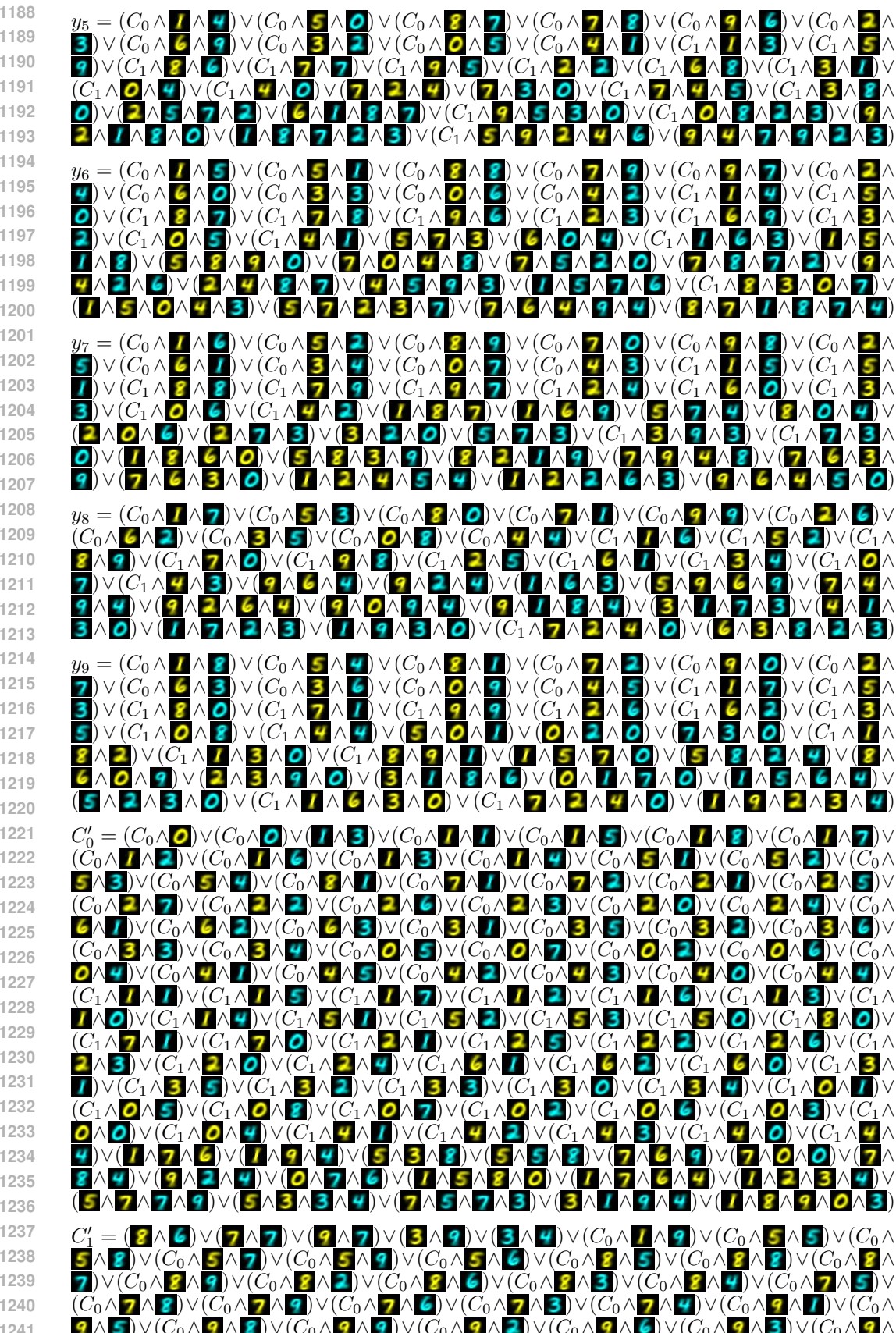

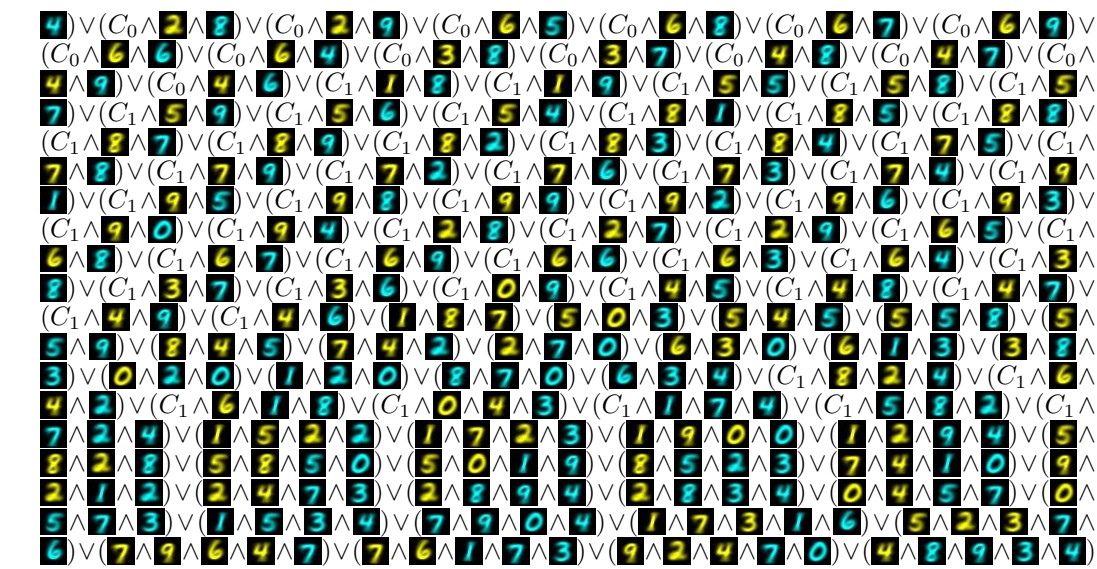

