# OpenReview forum: "Learning Arbitrary Logical Formula as a Sparse Neural Network Module"
_ICLR.cc/2025/Conference — Submitted to ICLR 2025_

### Official Review · Reviewer_a1S3 · 2024-10-28

**Soundness:** 2
**Presentation:** 1
**Contribution:** 3
**Rating:** 5
**Confidence:** 2

**Summary:**

The paper proposed a general framework of network modules that explicitly equate a logical formula after the convergence of another neural network. The model is end-to-end differentiable, and training all modules from scratch achieving joint convergence in a single run, and explicitly learning arbitrary logical formulas (within limited complexity) with its symbolic module.

However, I think this work is borderline rejected because of the following reasons:
1. Some notations are not defined clearly and are used wrong.
2. The structure of the manuscript is too messy to determine which part is the novelty and which is the preliminaries.

**Strengths:**

The authors conduct experiments with multiple settings to learn rules from a CNN model. The authors open their source code for the reference.

**Weaknesses:**

1. The structure of the paper is not clear to present which part is the novel contribution and which part is the preliminary work. For example, Sections 2.2.1 and 2.2.2 describe the Differentiable Neural Logic Network (dNL) Deep Symbolic Learning (DSL), these statements should be listed together in Preliminaries, not the contribution section. Furthermore, the terminology in Sections 2.2.1 and 2.2.2 should have citations.
2. Some formats do not fit the standard such as ‘define in 6’ in line 156 and ‘16 is also’ in line 241.
3. Some definitions are not very clear such as the ‘binary data’ in 103.
4. The authors only compare performance with Deep Symbolic Learning (DSL) proposed by Daniele et al., 2022. I think there should be more benchmarks to prove the performance of the proposed methods experimentally.

**Questions:**

1. There is no formal definition of ϵ-greedy policy in line 171, page 4.
2. There is no formal definition or citation for Gödel t-(co)norm in line 215 page 4.
3. The input of the softmax function is a vector. However, the input of the softmax function defined in Equation (9) line 173 page 4 is a real number, which is hard to be understood by readers.

---

> ### Author Response · Authors · 2024-11-25
>
> Thank you for your reviews and constructive comments. We acknowledge the points raised and will incorporate your suggestions in a future revision. We try to answer your questions below.
>
> Q1. There is no formal definition of ϵ-greedy policy in line 171, page 4.
>
> Line 178 says "For each classiffer, its predicted class is chosen with probability $1 − \epsilon1$, and a random class is chosen with probability $\epsilon1$." We will clarify this as the definition in a future revision.
>
> Q2. There is no formal definition or citation for Gödel t-(co)norm in line 215 page 4.
>
> The Gödel t-(co)norm is an important conception in fuzzy logic that is sometimes used without citations (but not always). We agree that adding a citation will make it clearer.
>
> Q3. The input of the softmax function is a vector. However, the input of the softmax function defined in Equation (9) line 173 page 4 is a real number, which is hard to be understood by readers.
>
> Line 176 says "$\softmax_j$ means applying softmax on the $j$ dimension." It is implemented as "torch.softmax(x, dim=1)" so this expression directly describes the code and might indeed be mathematically imprecise.

---

> > ### Comment · Reviewer_a1S3 · 2024-12-02
> >
> > The revision is improved in the aspect of self-contained. However, some concerns listed in Weaknesses have not been addressed. Hence, I will keep my score.

---

### Official Review · Reviewer_RJsP · 2024-11-01

**Soundness:** 1
**Presentation:** 1
**Contribution:** 2
**Rating:** 3
**Confidence:** 5

**Summary:**

The authors study three variants of differentiable inductive logic programming to both learn end-to-end neural classifiers and rules in a NeSy predictor setup. The architecture is based on a form of fuzzy operators with added stochasticity.

**Strengths:**

The task of learning both rules and classifiers is quite challenging, and the addition of noise via the Concrete distribution could be a good solution to this.

**Weaknesses:**

Currently, the paper is hard to follow, with a lot of abbrevations (it is difficult to remember what LFL-Type1-3 are!). The problem setup is not well-defined. I could infer it from my background, but this should be clear. I found it difficult to follow exactly how the 5 models (dNL, DSL and LFL-Type-i) fit in. The different LFL types are poorly motivated, and just are defined separately. Finally, the paper misses some critical related work, limiting the novelty of the paper.

The paper is limited in evaluation. It is only evaluated on a single task (MNIST Addition) and has a major problem in dataset setup which I will discuss below. Some different datasets, ideally also with different visual datasets than MNIST, would help with convincing that this model can properly learn.

**Questions:**

- The paper is missing several existing works on learning end-to-end learning: [1-5], of which [1-4] also learn proper logic rules. Furthermore, [4-6] are papers that also use an MLP loss / 'gradient shortcut' to (pre-)train the classifiers, which is critical to getting these systems to work. This is therefore not a new trick.
- I understand the relation to EQL, and it's good to cite it, but outside of inspiration, these two methods are not that directly related? It seems like LFL is more like differentiable inductive logic programming in the sense of [7] than symbolic regression.
- The footnote 2 is not correct: Scaling these weights does more than scaling the learning rate as it changes how close to 0.5 the average values of g(w_i) are. This means the derivatives of other parts of your system will change.
- The 'LFL' framework is not clearly and formally defined. What are the constraints on the different choices of the functions?
- The motivation for the linear reconstruction model is not clear to me. Is this purely for debugging?
- The data in MNIST Addition is not binary, so why is a binary cross-entropy loss used?
- What is the motivation for the label loss?
- Figure 3 can be trimmed significantly: A single one of these 4 images should suffice by just denoting what is optional.
- The architecture for multi-digit MNIST Addition in DSL (which I think is copied here) is highly specific to this task - How would this extend to different tasks?
- An experiment on data different than MNIST would be nice - MNIST is easily clusterable, making the MLP shortcut a useful signal, but this isn't the case for all data.
- The dataset setup for MNIST Addition is not correct and uses infinite data, while the canonical setup in the community is to take the training images, and randomly create a partition of them for different sums, giving finite data. The dataset is then about 60.000/2N in size - making the problem harder to train for larger multi-digit problems. This should be fixed.
- The relevance of experiment 3.1.1 could be clearer
- Experiment 3.2: I don't understand this result. There is nothing wrong with different training vs inference behaviour and this is standard in many DL layers. The label on Figure 4.d is unclear: This is _test_ accuracy under _training_ mode, right?

Minor
- The writing contains some typos. Eg: 052: Archived by - achieved by, 519: Conclution -> conclusion
- Be careful with quotes '', they should be different when opening


[1] Li, Zenan, et al. "Neuro-symbolic learning yielding logical constraints." Advances in Neural Information Processing Systems 36 (2024).

[2] Wang, Po-Wei, et al. "Satnet: Bridging deep learning and logical reasoning using a differentiable satisfiability solver." International Conference on Machine Learning. PMLR, 2019.

[3] Cunnington, Daniel, et al. "The role of foundation models in neuro-symbolic learning and reasoning." International Conference on Neural-Symbolic Learning and Reasoning. Cham: Springer Nature Switzerland, 2024.

[4] Aspis, Yaniv, et al. "Embed2Rule Scalable Neuro-Symbolic Learning via Latent Space Weak-Labelling." International Conference on Neural-Symbolic Learning and Reasoning. Cham: Springer Nature Switzerland, 2024.

[5] Daniele, Alessandro, et al. "Simple and Effective Transfer Learning for Neuro-Symbolic Integration." International Conference on Neural-Symbolic Learning and Reasoning. Cham: Springer Nature Switzerland, 2024.

[6] Aspis, Yaniv, et al. "Embed2Sym: scalable neuro-symbolic reasoning via clustered embeddings." Kern-Isberner G, Lakemeyer G, Meyer T, editors. Proceedings of the 19th International Conference on Principles of Knowledge Representation and Reasoning (KR2022); 2022 Jul 31-Aug 5; Haifa, Israel.[California]: IJCAI Organization; 2022. p. 421-31.. International Joint Conferences on Artificial Intelligence Organization, 2022.

[7] Evans, Richard, and Edward Grefenstette. "Learning explanatory rules from noisy data." Journal of Artificial Intelligence Research 61 (2018): 1-64.

---

> ### Author Response · Authors · 2024-11-25
>
> Thank you for your reviews and constructive comments. We acknowledge the points raised and will incorporate your suggestions in a future revision. We try to answer your questions below.
>
> Q1. The paper is missing several existing works on learning end-to-end learning: [1-5], of which [1-4] also learn proper logic rules. Furthermore, [4-6] are papers that also use an MLP loss / 'gradient shortcut' to (pre-)train the classifiers, which is critical to getting these systems to work. This is therefore not a new trick.
>
> They are indeed previous works in the NeSy subfield. In the current version we referenced mainly NeSy predictors that take the MNIST Arithmetic tasks as the baseline, so we skipped [1] and [2] in the related works section. We added the tricks mainly because we needed them to make the system converge in a single run rather than proposing the tricks as major contributions, so we will probably just refer to these tricks as "tricks necessary for joint convergence" in a future revision. [4] and [6] extract symbolic rules from pretrained networks by clustering the learned embeddings, while [5] proposes a 2-stage training technique that works on general NeSy predictors by pretraining the CNNs with the symbolic component replaced by MLP. We agree that it's good to add them to the related works in a future revision.
>
> Q2. I understand the relation to EQL, and it's good to cite it, but outside of inspiration, these two methods are not that directly related? It seems like LFL is more like differentiable inductive logic programming in the sense of [7] than symbolic regression.
>
> This work actually take some designs from an EQL work [8] which includes a Symbolic Regression version of the MNIST Sum task. The idea of extracting symbolic expression with SymPy comes directly from it's implementation, and it uses a L0 regularization method based on the Binary Concrete distribution that brought us to the distribution. We will clarify them in a future revision.
>
> The method is indeed related to inductive logic programming, as dNL is originally proposed as an ILP method. We experiment them mainly on NeSy predictors (rather than ILP tasks) because the NeSy predictor tasks are more helpful for our future plan of integrating LFLs into MBRL agents, as we mentioned in the future work section.
>
> Q3. The footnote 2 is not correct: Scaling these weights does more than scaling the learning rate as it changes how close to 0.5 the average values of g(w_i) are. This means the derivatives of other parts of your system will change.
>
> You are right. They would be equivalent only if the initial values of the trainable parameters are also scaled together with the learning rate and the optimizers always update the parameters proportionally to the learning rates. Generally, in this work we scale the learning rates instead of scaling the hyperparameters following DSL for consistency in implementation.
>
> [1] Li, Zenan, et al. "Neuro-symbolic learning yielding logical constraints." Advances in Neural Information Processing Systems 36 (2024).
>
> [2] Wang, Po-Wei, et al. "Satnet: Bridging deep learning and logical reasoning using a differentiable satisfiability solver." International Conference on Machine Learning. PMLR, 2019.
>
> [3] Cunnington, Daniel, et al. "The role of foundation models in neuro-symbolic learning and reasoning." International Conference on Neural-Symbolic Learning and Reasoning. Cham: Springer Nature Switzerland, 2024.
>
> [4] Aspis, Yaniv, et al. "Embed2Rule Scalable Neuro-Symbolic Learning via Latent Space Weak-Labelling." International Conference on Neural-Symbolic Learning and Reasoning. Cham: Springer Nature Switzerland, 2024.
>
> [5] Daniele, Alessandro, et al. "Simple and Effective Transfer Learning for Neuro-Symbolic Integration." International Conference on Neural-Symbolic Learning and Reasoning. Cham: Springer Nature Switzerland, 2024.
>
> [6] Aspis, Yaniv, et al. "Embed2Sym: scalable neuro-symbolic reasoning via clustered embeddings." Kern-Isberner G, Lakemeyer G, Meyer T, editors. Proceedings of the 19th International Conference on Principles of Knowledge Representation and Reasoning (KR2022); 2022 Jul 31-Aug 5; Haifa, Israel.[California]: IJCAI Organization; 2022. p. 421-31.. International Joint Conferences on Artificial Intelligence Organization, 2022.
>
> [7] Evans, Richard, and Edward Grefenstette. "Learning explanatory rules from noisy data." Journal of Artificial Intelligence Research 61 (2018): 1-64.
>
> [8] Kim S, Lu P Y, Mukherjee S, et al. Integration of neural network-based symbolic regression in deep learning for scientific discovery[J]. IEEE transactions on neural networks and learning systems, 2020, 32(9): 4166-4177.

---

> ### Author Response · Authors · 2024-11-25
>
> Q4. The 'LFL' framework is not clearly and formally defined. What are the constraints on the different choices of the functions?
>
> The system may work as long as the t-(co)norms and $g$ are differentiable. At line 140 of the paper there is "An LFL network can be constructed by arbitrarily combining these differentiable neurons.", which can be considered as the definition. We agree that it should be clarified in a future revision.
>
> Q5. The data in MNIST Addition is not binary, so why is a binary cross-entropy loss used?
>
> The binary cross-entropy loss is used because the output labels are binary. The networks need to predict one-hot binary labels from 19 choices, representing that the sum is 0-18.
>
> Q6. What is the motivation for the label loss?
>
> The networks would not converge without it. An MLP is a universal approximator for real-valued functions, so the MLP can predict the output using less symbols than the LFL module, and this constraint forces the MLP to use all the classification symbols for prediction. Without the loss term, the MLP would probably use only 9 symbols from each MNIST image to predict the sum, for example.
>
> Q7. Figure 3 can be trimmed significantly: A single one of these 4 images should suffice by just denoting what is optional.
>
> We agree.
>
> Q8. The architecture for multi-digit MNIST Addition in DSL (which I think is copied here) is highly specific to this task - How would this extend to different tasks?
>
> For recursive tasks other than multi-digit MNIST Addition, the input would be replaced by the actual inputs and the carry "c" would be replaced by the recursive values useful for the task. The DSL paper provided very formal definitions on this, maybe we can clarify it more briefly in a future revision.
>
> Q9. An experiment on data different than MNIST would be nice - MNIST is easily clusterable, making the MLP shortcut a useful signal, but this isn't the case for all data.
>
> We actually experimented on CIFAR Sum during the past few days. The proposed networks do converge on CIFAR Sum training data, but the ResNet18 used for CIFAR Sum overfits on CIFAT10 training images, causing the test set accuracy to be lower than those reported in [1].
>
> Q10. The dataset setup for MNIST Addition is not correct and uses infinite data, while the canonical setup in the community is to take the training images, and randomly create a partition of them for different sums, giving finite data. The dataset is then about 60.000/2N in size - making the problem harder to train for larger multi-digit problems. This should be fixed.
>
> We agree.
>
> Q11. The relevance of experiment 3.1.1 could be clearer.
>
> We want to conduct a simple test on whether the LFL-Type1 module works with more than 2 layers on binary input/output. That will be clarified in a future revision.
>
> Q12. Experiment 3.2: I don't understand this result. There is nothing wrong with different training vs inference behaviour and this is standard in many DL layers. The label on Figure 4.d is unclear: This is test accuracy under training mode, right?
>
> That's training accuracy under training mode, but there would be little difference between training and test accuracies on MNIST Arithmetic tasks since MNIST images are easy to classify. We will probably remove Experiment 3.2 and LFL-Type2 in a future revision because there is nothing wrong with different training vs inference behaviour and this is standard in many DL layers.
>
> [1] Aspis, Yaniv, et al. "Embed2Rule Scalable Neuro-Symbolic Learning via Latent Space Weak-Labelling." International Conference on Neural-Symbolic Learning and Reasoning. Cham: Springer Nature Switzerland, 2024.

---

> > ### Comment · Reviewer_RJsP · 2024-11-25
> >
> > I thank the authors for their extensive comments and openness to feedback. I will retain my score, since the number of changes required for this revision is too great for the camera-ready. However, I wish the authors all the luck with the future vision.
> >
> > > In the current version we referenced mainly NeSy predictors that take the MNIST Arithmetic tasks as the baseline
> >
> > I would also recommend trying to tackle other tasks than (MNIST) Arithmetic. This task is to a large degree 'solved' by the community in various variations. Showing that LFL works also on other tasks will improve the confidence readers will have in LFL. CIFAR arithmetic is definitely more challenging, but since LFL is about rule learning, different types of rules would also be important.
> >
> > > The networks need to predict one-hot binary labels from 19 choices, representing that the sum is 0-18.
> >
> > The proper loss for one-hot labels with 19 choices is a softmax-cross entropy, though, since these options are mutually exclusive. But maybe I am missing something in the setup.

---

> > > ### Author Response · Authors · 2024-11-25
> > >
> > > Although these options are mutually exclusive in the classification labels, LFL-Type1 and LFL-Type3's outputs are not mutually exclusive, so we think it would make more sense to train them with binary cross-entropy loss. DSL also used the binary cross-entropy loss because its outputs are non-zero only for the chosen symbol, making the softmax-cross entropy unusable.

---

### Official Review · Reviewer_Qb9Q · 2024-11-02

**Soundness:** 3
**Presentation:** 4
**Contribution:** 3
**Rating:** 8
**Confidence:** 3

**Summary:**

The paper introduces the LFL (Logical Formula Learner), an architecture that entails and expands prior work, in particular DSL (Deep Symbolic Learning) and dNL (differentiable Logic Network) . Both prior works feature learnable logical representations. LFL proposes an architecture that includes both DSL and dNL, but within LFL the user can choose more or less representational freedom. This architecture also includes MLPs as loss shortcuts and several loss terms to control convergence.

Evaluation is done on multi-digit MNist addition, MNist-addition and on 3-layer logical formula.

**Strengths:**

Very well written, clearly presented.

The problem is interesting, as learning logical formulae opens many doors for NeSy learning.

The approach seems to be mostly original: The foundations (DSL and dNL) were clearly acknowledged and delineated from original contributions by the authors.

**Weaknesses:**

Figure 4(d) seems a bit dishonest. Apparently DSL's poor performance is down to solely training mode settings. This should be presented more clearly and transparently, e.g. by showing test performance.

The paper could use a comparison to some of the competing branches of related work, e.g. differentiable/neural ILP.

Various typos:
Typo achieved / archived bottom of page 1.
Fig 3's caption: references to "such as [9] and [16]" should probably read "such as Equations [9] and [16]".
"In these networks we also constrain**t**" -> "In these networks we also constrain".

**Questions:**

It is not completely clear to me to what degree other approaches rely on the same amount of practical/engineering tricks to achieve proper convergence. I'm under the impression that e.g. ILP does not need this as heavily, but I also have to admit to not being an expert on logic formula learning.

---

> ### Author Response · Authors · 2024-11-25
>
> Thank you for your reviews and constructive comments. We acknowledge the points raised and will incorporate your suggestions in a future revision. We try to answer your questions below.
>
> Q1. It is not completely clear to me to what degree other approaches rely on the same amount of practical/engineering tricks to achieve proper convergence. I'm under the impression that e.g. ILP does not need this as heavily, but I also have to admit to not being an expert on logic formula learning.
>
> [1] also reports that "Deep Symbolic Learning (DSL) struggles to converge when dealing with problems with large amounts of latent symbols due to the large hypothesis space it needs to explore." In general these differentiable logic modules are far less effective than MLPs on passing information back into previous modules, even though they may work well on symbolic training data provided in ILP tasks.
>
> [1] Simple and Effective Transfer Learning for Neuro-Symbolic Integration. Daniele et al. 2024

---

> > ### Comment · Reviewer_Qb9Q · 2024-11-26
> >
> > Thanks for the clarification.
> >
> > Looking at the other reviews, I seem to have found your paper more readable than the others, and I stand by that assessment.

---

### Official Review · Reviewer_bBsq · 2024-11-04

**Soundness:** 2
**Presentation:** 2
**Contribution:** 2
**Rating:** 3
**Confidence:** 3

**Summary:**

This work proposes a logic formula learner framework, which comprises of three components: LFL-Type1 learns arbitrary logical formula, LFL-Type2 learns a look-up table, and LFL-Type3 has combinatorial search freedom between them. This frame work is end-to-end differentiable, can converge in a single run, and can learn arbitrary logic formula with the symbolic module.

**Strengths:**

Originality: 2.5/5

Compared to the existing work DSL, this approach offers the ability to explicitly learn the logic equation. It introduces a Concrete distribution alongside the Godel t-(co)norm proposed in DSL, which adds noise to the soft symbolic values.

Quality: 2/5

Pros: The context and mathematical formulations are well presented.

Cons: Scalability is a significant concern, as the number of neurons correlates with the number of internal states. While DSL has demonstrated scalability up to a 1000-digit sum, this work does not address its own scalability. How complex can this approach effectively scale? In the experimental section, the random seed is fixed at 42, which is unexplained.

Clarity: 2/5

Pros: The introduction provides a clear and convincing narrative.

Cons: The figures do not effectively support comprehension. For example, while Figure 1 illustrates the different outcomes of EQL and AFL, fully understanding it requires a detailed study of both paradigms. Figure 3 is even more challenging, as the components are not well explained in the caption, and there are many variants with minor changes. Figure 4 lacks sufficient details for understanding the experiments and data fully.

Significance: 2/5

Scalability is the primary limitation impacting the influence of this work on the neurosymbolic community.

**Weaknesses:**

See strengths above.

**Questions:**

1.  How complex can this approach effectively scale to?
2. In the experimental section, the random seed is fixed at 42, which is unexplained.

---

> ### Author Response · Authors · 2024-11-25
>
> Thank you for your reviews and constructive comments. We acknowledge the points raised and will incorporate your suggestions in a future revision. We try to answer your questions below.
>
> Q1. How complex can this approach effectively scale to?
>
> This approach is just as scalable as DSL since we just replace the differentiable logic module and add some other network components. In the current version of this paper we experimented with 128-digit sum (we used $n_{digits}$ to denote the length), and scaling it up to 1000-digit sum will just get the same result.
>
> Q2. In the experimental section, the random seed is fixed at 42, which is unexplained.
>
> Random seed 42 is commonly used in machine learning researches or competitions as there is some story behind it (an explanation can be seen at https://medium.com/geekculture/the-story-behind-random-seed-42-in-machine-learning-b838c4ac290a). We use this seed to show that we didn't cherry pick the seeds.

---

> > ### Comment · Reviewer_bBsq · 2024-11-28
> >
> > For Q2, I would suggest to run multiple runs of experiments, and obtain the error bar for your result.
> >
> > Although the rebuttal addressed some of my concerns, the clarity is a significant issue, and I would like to maintain my rating.

---

### Official Review · Reviewer_cUik · 2024-11-10

**Soundness:** 1
**Presentation:** 1
**Contribution:** 3
**Rating:** 3
**Confidence:** 3

**Summary:**

Summary:
The authors propose a Neuro-Symbolic(NeSy) method that can learn rules from perception in an end-to-end fashion. The paper aims to improve upon recent lines of work in this direction, and proposes new designs and symbol selection strategies, with different levels of noise and distributions. The paper also proposes a gradient shortcut strategy to improve the training and convergence. The paper also show experiments on synthetic MNIST based NeSy benchmarks --- widely used to test NeSy frameworks.

**Strengths:**

- The general idea of having membership functions for symbols is interesting, and is potentially an elegant abstraction of ideas from DSL.
- The idea of learning rules from perception in general is very interesting, and is of significant importance in NeSy.

**Weaknesses:**

- Clarity: I think the paper is very unclear. This can be significantly improved by adding a running example, and potentially moving the comparison (3.1) and parts of section 4 to the Appendix. However, the general writing of related works and introduction is not quite clear. I would suggest the authors to arrive at the main message of the paper at earlier stage in the introduction.

- Section 2.1 is well written, and well-motivated. However, 2.3 and 2.3 (in my understanding the main contributions of the paper) are not clearly presented. It is not clear why Concrete distributions help?

- Figure 3's explanation is quite unclear and imprecise.

- I am not sure, how novel the idea of gradient shortcuts is --- see questions.

**Questions:**

- Please summarily explain why concrete distributions help compared to $\epsilon$-greedy?
- How does your new trick compare to [1]?

[1] Simple and Effective Transfer Learning for Neuro-Symbolic Integration. Daniele et al. 2024

---

> ### Author Response · Authors · 2024-11-25
>
> Thank you for your reviews and constructive comments. We acknowledge the points raised and will incorporate your suggestions in a future revision. We try to answer your questions below.
>
> Q1. Please summarily explain why concrete distributions help compared to $\epsilon$-greedy?
>
> The $\epsilon$-greedy cannot be applied to sigmoid-like memberships like those in dNL, so we turn to the Binary Concrete distribution to add noise to the memberships in LFL-Type1 and LFL-Type3 (they wouldn't work without noise because the Gödel t-(co)norm backpropagates sparse gradients).  Then we tried to use the softmax-like Concrete distribution to produce one variation with the same combinatorial search freedom as DSL (LFL-Type2) and find it works as well. The only advantage that LFL-Type2 has over DSL is that it has no gap between training and inference behaviors; we now agree that the advantage seems too minor to be proposed. We may remove LFL-Type2 in a future revision of this paper.
>
> Q2. How does your new trick compare to [1]?
>
> [1] is a recent work whose peer-reviewed version was published in September 2024. It proposes a training procedure that pretrains a NeSy model with its symbolic component replaced by an MLP before training the NeSy predictor. So it's a two-stage training strategy that applies to all NeSy models, while our tricks used in this work applies to end-to-end differentiable NeSy predictors and train the whole system from scratch in a single run.
>
> [1] Simple and Effective Transfer Learning for Neuro-Symbolic Integration. Daniele et al. 2024

---

> > ### Comment · Reviewer_cUik · 2024-11-26
> > **Thanks for the response**
> >
> > I thank the authors for addressing some of the concerns raised in the review. However, I think clarity has been a major concern for me. And I find it hard that it can be easily fixed within the horizon of the review period. I will keep my score.

---

### Meta-Review · Area_Chair_8ajF · 2024-12-20

**Metareview:**

The paper introduces the Logical Formula Learner (LFL). LFL is based on DSL (Deep Symbolic Learning) and dNL (differentiable Logic Network) and "mixes" Goedel t-(co)norm and the Concrete distribution together. While overall the reviewers (and I) agree that the question tackled and the direction taken are interesting, ramework which uses LLMs to create a multi-agent system, used to attempt Kaggle problems. They use a "phase-based" multi-agent approach, together with a library of hand-crafted ML tools, and extensive hand-crafted unit-tests tailored to the Kaggle problems. The main downside is the focus on MNIST Addition experiments (scaling experiment should be added, as just arguing it is the same as X is not the same as simply running scaling experiments; this would also be a good place to compare to other neurosymbolic approaches) and the weak presentation. While I agree that the presentation has to be improved, the major downside is that only one domain/task is considered. Actually, the paper does not really explain the experimental methodology.  How do you select the task? Why no other major tasks or domains? You do not perform statistical tests to assess whether any observed differences in performance are significant, and doing so requires several reruns with different seeds. To summarize, this is an interesting direction, but the paper is not ready for publication at ICLR. Please note that the overall judgment should not be taken as a statement regarding the usefulness of your research.

**Additional Comments On Reviewer Discussion:**

The discussion arose from problems and questions raised in the reviews. One of the topics was the clarity of the text, though this was not the main argument for the overall decision. The one small discussion about MNIST and missing other domains was more important for the overall discussion.

---

### Decision · Program_Chairs · 2025-01-22

Reject